# Senescent cells limit p53 activity via multiple mechanisms to remain viable

Ines Sturmlechner [1,2], Chance C. Sine[1], Karthik B. Jeganathan[1], Cheng Zhang[3], Raul O. Fierro Velasco [1], Darren J. Baker [1,4], Hu Li [3] & Jan M. van Deursen [1,4 ✉]

Super-enhancers regulate genes with important functions in processes that are cell type-specific or define cell identity. Mouse embryonic fibroblasts establish 40 senescence-associated super-enhancers regardless of how they become senescent, with 50 activated genes located in the vicinity of these enhancers. Here we show, through gene knockdown and analysis of three core biological properties of senescent cells that a relatively large number of senescence-associated super-enhancer-regulated genes promote survival of senescent mouse embryonic fibroblasts. Of these, *Mdm2, Rnase4*, and *Ang* act by suppressing p53-mediated apoptosis through various mechanisms that are also engaged in response to DNA damage. *MDM2* and *RNASE4* transcription is also elevated in human senescent fibroblasts to restrain p53 and promote survival. These insights identify key survival mechanisms of senescent cells and provide molecular entry points for the development of targeted therapeutics that eliminate senescent cells at sites of pathology.

---

[1] Department of Pediatric and Adolescent Medicine, Mayo Clinic, Rochester, MN, USA. [2] Department of Pediatrics, Molecular Genetics Section, University of Groningen, University Medical Center Groningen, Groningen, The Netherlands. [3] Department of Molecular Pharmacology and Experimental Therapeutics, Mayo Clinic, Rochester, MN, USA. [4] Department of Biochemistry and Molecular Biology, Mayo Clinic, Rochester, MN, USA. ✉email: janvandeursen2@gmail.com

The cellular senescence program induces a stable cell cycle arrest in response to various types of cellular stresses through a complex interplay between the Rb and p53 tumor suppressor pathways[1,2]. Although entry into the senescent state results from cellular stress, it is important that senescent cells (SNCs) stay alive to exert beneficial effects in wound healing[3,4], tissue regeneration[5–7], and tumor suppressive immunosurveillance[8,9]. A key SNC non-autonomous characteristic termed the senescence-associated secretory phenotype (SASP) is thought to underlie many of these physiological as well as non-physiological aspects of SNCs. Although the composition of the SASP varies depending on the senescent-cell inducer and the cell type, it typically includes factors modulating growth, the immune system, or the extracellular matrix[10]. Additionally, SNCs have been shown to accumulate with aging and in association with various age-related disorders[11–15]. In these contexts, SNCs are thought to be largely detrimental, driving aspects of tissue degeneration and disease pathology[16,17].

Increasing evidence indicates that SNCs are a heterogenous collection of cells with multifaceted features. The type of senescence-inducing stimulus, cell type and species, as well as temporal kinetics can all contribute to SNC heterogeneity, their properties, and evolution[18,19]. Such SNC complexity has made it challenging to identify universal markers of senescence, which has hindered in-depth studies on SNCs from human tissues and organs and the development of therapeutics that selectively target SNCs with pathological properties. Unbiased genome-wide approaches such as transcriptomic profiling via RNA-sequencing have underscored SNC transcriptional heterogeneity originating from cell types and stressors[20–22]. Transcriptional profiling typically identifies hundreds to thousands of differentially expressed genes (DEGs) upon senescence due to a particular stressor, but comparison across stressors and cell types yields relatively few common DEGs[20,21]. Additionally, marked cell-to-cell gene expression variability exists in SNC cultures in vitro derived from the same cell type induced by the same stressor[23,24], further complicating the comprehensive identification and study of SNCs. Technological advances such as single-cell RNA-sequencing in combination with comprehensive open-source databases including SeneQuest[25] or CellAge[26] will not only simplify integration of SNC transcriptomic data, but also facilitate the identification of widely applicable senescence markers.

Although unbiased transcriptomic techniques can be potent hypothesis-generating approaches, methods to decipher epigenetic complexities of cells during homeostasis and diseases, including SNCs, have emerged as powerful tools[27,28]. In particular, the concept of super-enhancers (SEs) controlling cell identity emerged as an important model during the past decade. SEs are large enhancers, binding numerous transcription factors and co-activators, and are rich in various chromatin modifications[29]. Importantly, SEs govern cell identity and orchestrate cell type-selective gene expression patterns[30,31], as has been demonstrated in a variety of cell and tissue types during cell differentiation[32–34], as well as in disease contexts, including multiple types of cancer[35,36], atherosclerosis[37] and auto-immune disease[38]. The discovery that disease-relevant cells rely on SEs has led to an exciting therapeutic concept in which small-molecule inhibitors targeting SE co-activators such as BRD4 interfere with SE function. This approach has shown promise in mouse models and clinic trials on cancer patients have been initiated[36]. Furthermore, the identification of crucial disease-relevant SEs has emerged as a potent tool to identify SE target genes that drive cellular key properties causative to diseases. Targeting such SE-controlled genes or associated molecular effector pathways holds great therapeutic promise. For example, multiple drug-targetable oncogenic genes and pathways were discovered by identifying SEs gained in chemotherapy-resistant brain tumors (ependymomas)[39].

Changes in the SE landscape upon transition into cellular senescence due to replicative exhaustion were previously identified and led to the discovery of p300 as a mediator of the senescence program[40]. Additionally, SEs of oncogene-induced (OI) SNCs were characterized as vital to SASP factor expression, and inhibition of the SE co-activator BRD4 showed impaired SASP production and immunosurveillance in mice[41]. While these studies unearthed important principles of senescence-associated SEs (SASEs) per se, in-depth studies on SASE target genes and their significance to SNC identity and key properties is lacking. Addressing this gap in knowledge holds not only promise to deepen our mechanistic understanding for how SNCs stay growth arrested, alive, and able to produce the SASP, but also opens the possibility to unravel SNC vulnerabilities that may be leveraged for therapeutic benefit to eliminate disease-relevant SNCs or neutralize their detrimental properties.

In a recent proof-of-concept study, we screened for SASE-controlled genes that are highly conserved across species, cell types, and senescence-inducing stressors, which led to the identification of three such genes, *Hivep2*, *Rtn2* and *p21* (*Cdkn2a*)[9]. In-depth characterization of *p21* revealed that this p53 target gene activates retinoblastoma protein (Rb)-dependent transcription at select gene promoters to produce a bioactive secretome, referred to as the p21-associated secretory phenotype (PASP), that places stressed cells under immunosurveillance as part of a biological timer mechanism that controls cell fate. In the current study, we further pursued the idea that SASE-associated genes have particularly important functions in SNCs by focusing on 50 previously identified SASE-controlled genes in mouse embryonic fibroblasts (MEFs) that are conserved across three distinct senescence-inducing stressors: high-dose γ-irradiation (IR), extensive replication (REP) or oncogenic signaling by KRAS$^{G12V}$ (OI)[9]. We systematically examined the functional significance to three core properties of SNCs, cell cycle arrest, the senescence-associated secretory phenotype (SASP), and survival. We found that a high proportion of SASE genes stimulate the expression of SASP factors or promote SNC survival, and that preventing p53-mediated apoptosis is a frequent mechanistic theme among SASE genes required for survival, providing entry points for the development of targeted senolytic strategies.

## Results

**SASE genes function in core SNC properties but primarily in survival.** Previously, we identified 40 common SASEs in senescent MEFs via H3K27ac chromatin immunoprecipitation and sequencing (H3K27ac ChIP-seq)[9], which is considered a less stringent method for mapping super-enhancers than MED1 ChIP-seq[30]. We further reported that 50 transcriptionally activated genes are associated with these common SASEs (Supplementary Fig. 1)[9]. Of the SASE-associated genes, we selected a subset of 28 upregulated protein-coding genes for experimental follow-up studies. These genes are listed in Supplementary Fig. 1c. We first identified, for each SASE gene, two lentivirally-delivered shRNAs that substantially reduced transcript levels in IR-SNCs (Fig. 1a and Supplementary Fig. 2), and then used these to assess the impact of gene knock-down on IR-SNC survival, cell cycle arrest and core components of the SASP (Fig. 1a). Genes significantly impacting one or more properties of IR-SNCs were then further tested in REP- and OI-senescent MEFs to assess whether their effects were consistent regardless of the senescence-inducing stimulus.

We first evaluated the maintenance and stability of cell cycle arrest. Our subset of 28 SASE genes included two genes with established roles in cell cycle arrest, *p21* and *Rb*[42,43]. As expected, IR-SNCs depleted for *p21* or *Rb* entered S-phase as measured by

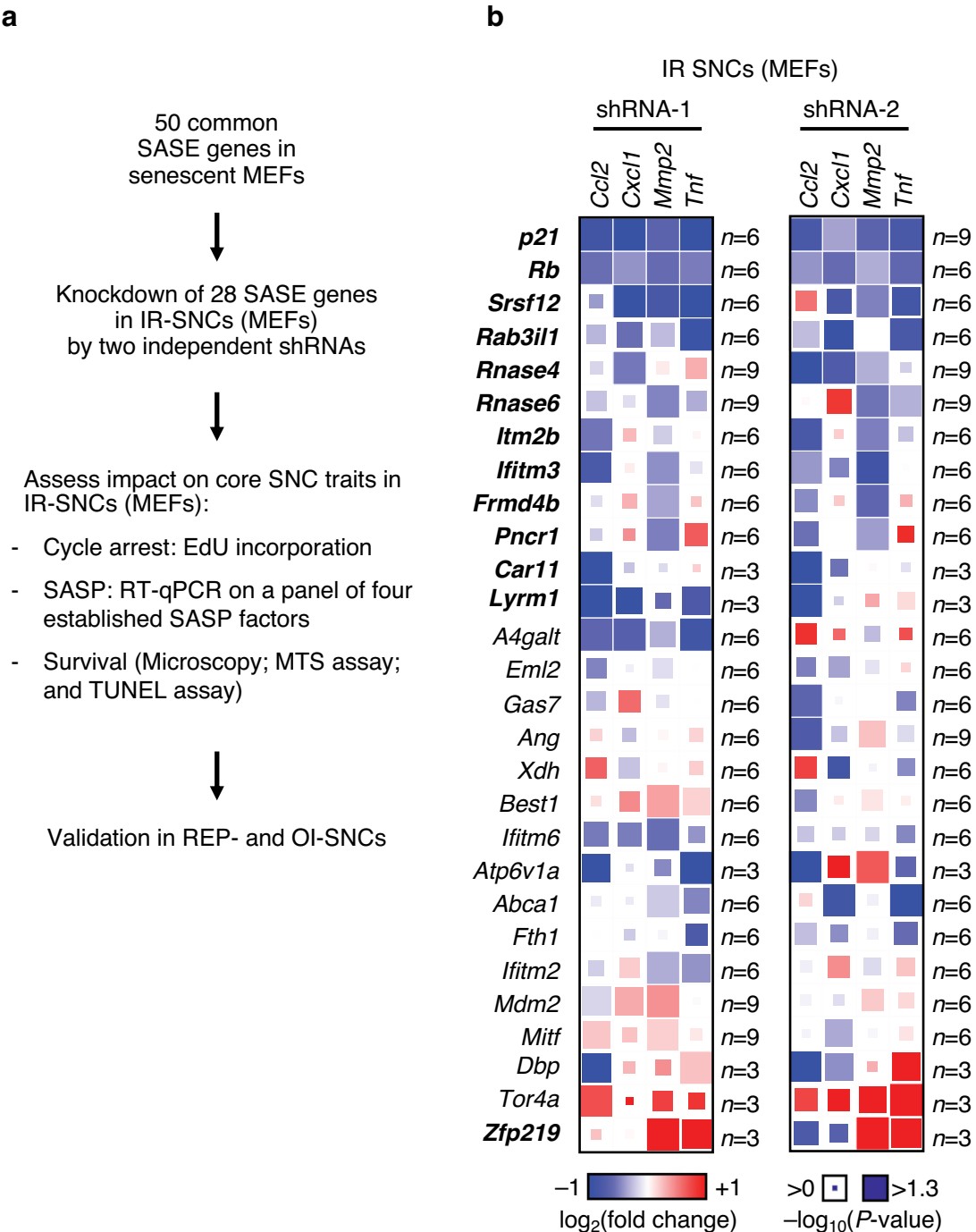

**a**

50 common SASE genes in senescent MEFs

↓

Knockdown of 28 SASE genes in IR-SNCs (MEFs) by two independent shRNAs

↓

Assess impact on core SNC traits in IR-SNCs (MEFs):

- Cycle arrest: EdU incorporation

- SASP: RT-qPCR on a panel of four established SASP factors

- Survival (Microscopy; MTS assay; and TUNEL assay)

↓

Validation in REP- and OI-SNCs

**Fig. 1 A subset of SASE genes controls the expression of select SASP factors. a** Strategy for the systematic characterization of a subset of 28 SASE genes (here defined as genes 50 kb up- or down-stream of a SASE and transcriptionally induced in SNCs). Involvement of individual SASE genes in three core properties of SNCs was first assessed in IR-SNCs. Significant results consistently obtained with two shRNAs were validated in REP- and OI-SNCs. **b** Heatmap depicting $\log_2$ fold changes of the expression of the indicated SASP factors in IR-SNCs three days after SASE gene knockdown as assessed by RT-qPCR. Expression levels of these factors in the same IR-SNC cultures infected *shScr*-virus were used to assess fold changes. Heatmap shows $\log_2$ fold changes as box color and statistical significance using paired, two-tailed *t*-tests as box size. Data represent means. *n* depict independent MEF lines that were pooled from 1 to 3 independent experiments in **b**. Source data are provided as a Source Data file.

5-ethynyl-2′-deoxyuridine (EdU) incorporation, but none of the other SASE genes increased EdU incorporation when depleted (Supplementary Fig. 3), indicating they are not required to keep SNCs in a non-proliferative state.

Next, we determined the extent to which SASE genes had key contributions to SASP gene expression. To this end, we depleted each SASE gene in IR-senescent MEFs, harvested RNA and performed RT-qPCR on a small panel of well-established SASP factors upregulated in senescent MEFs[10]. Individual depletion of 12 SASE genes, *Rb, Cdkn1a, Srsf12, Rab3il1, Rnase4, Rnase6, Itm2b, Ifitm3, Frmd4b, Pnrc1, Car11* and *Lyrm1* significantly reduced expression of at least one SASP factor with two independent shRNAs (Fig. 1b). By contrast, *Zfp219* increased the expression of SASP factors when depleted, suggesting that this

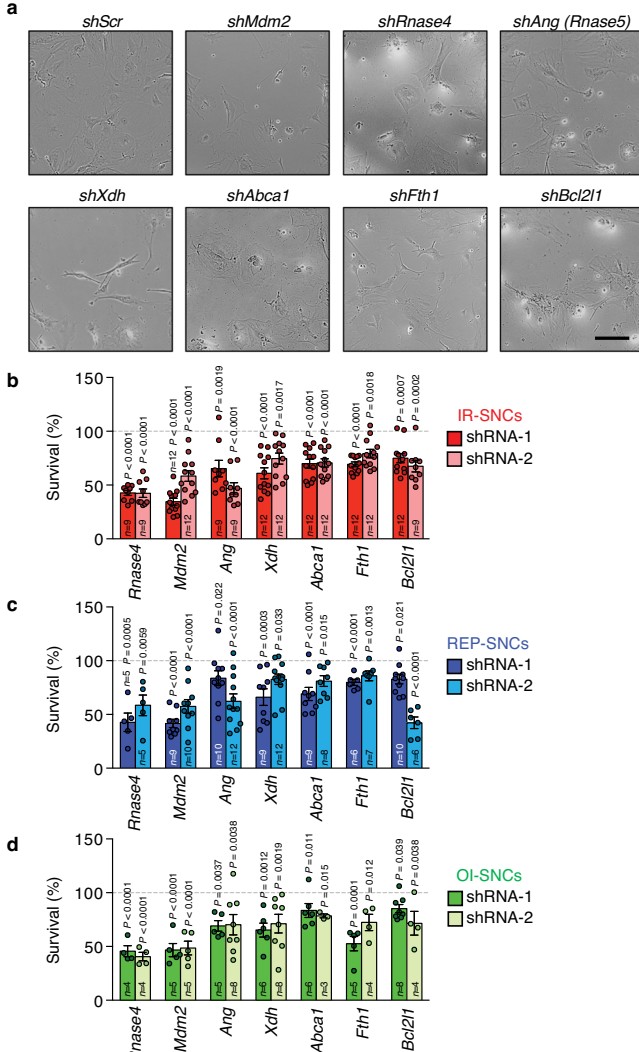

**Fig. 2 Six SASE genes are commonly required to sustain SNC survival.**
**a** Images of IR-senescent MEFs 6 days after shRNA-mediated knockdown of the indicated SASE genes. Images are representative of 3 independent experiments. **b–d** Survival of IR- **b**, REP- **c**, and OI-SNCs **d** at day 6 after SASE gene knockdown with two independent shRNAs per gene. Survival was measured by MTS assay. Comparisons were made to SNCs of same cultures infected with scrambled shRNA (shScr; 100% line). Scale bar, 200 μm. Data represent means ± SEM. n depict independent MEF lines that were pooled from 2 to 4 independent experiments. Statistics: one-way ANOVA with Sidak's correction in **b–d**. Source data are provided as a Source Data file.

SASE gene limits the extent to which certain SASP factors are upregulated in the senescent state. These data imply that genes associated with SASEs play a central role in sustaining the SASP and underscore that the SASP is established by an intricately regulated gene network.

Lastly, we determined the extent to which SASE gene depletion impacts SNC viability. As a positive control, we included *Bcl2l1* (encoding BCL-xL), a key target of the well-established senolytic agents ABT-737 and ABT-263[44,45]. Strikingly, a large proportion of SASE genes significantly reduced survival of IR-SNCs with two independent shRNAs (Fig. 2a, b, and Supplementary Fig. 4a, b). Of the 11 genes identified, six (*Mdm2*, *Rnase4*, *Ang* (*Rnase5*), *Abca1*, *Xdh*, and *Fth1*) also showed reduced survival in both REP- and OI-SNCs (Fig. 2c, d and Supplementary Fig. 4c, d). All these

SASE genes caused apoptosis when depleted in IR-SNCs as assessed by terminal deoxynucleotidyl transferase-mediated deoxyuridine triphosphate nick end labeling (TUNEL) assay (Fig. 3a). In all cases, with the exception of *Xdh*, apoptosis rates in SNCs were markedly higher than in their proliferating counterparts (Fig. 3b). The MTS and TUNEL assays both indicated that *Mdm2*, *Rnase4*, and *Ang* were at least as important for SNC survival as *Bcl2l1*. Collectively, these data suggested that SNCs depend on multiple SASE genes for sustained survival. We prioritized these genes for in-depth mechanistic studies because of their potential to inform targeted senolytic approaches for elimination of disease-associated SNCs in humans.

**Multiple SASE genes promote survival through inhibition of p53.** To explore how SNCs counteract apoptosis, we focused on p53 because two of the SASE genes promoting survival, *Mdm2* and *Ang*, inhibit p53, including its pro-apoptotic activities[46,47]. ANG is a less well-established p53 inhibitor than MDM2. It interacts with p53 and restricts its transcriptional activity, including transcription of pro-apoptotic target genes[48,49]. We hypothesized that SNCs activate both *Mdm2* and *Ang* expression in the senescent state to increase survival by limiting the extent of p53 activity. As a first step to test this idea, we depleted SASE genes that increase survival with or without co-depletion of p53. We included *Bcl2l1* in this analysis, as BCL-xL has also been demonstrated to bind and inhibit p53 to limit its pro-apoptotic function at mitochondria[50,51]. Indeed, *p53* knockdown in combination with *Mdm2*, *Ang*, or *Bcl2l1* knockdown restored SNC viability independently of the senescence-inducing stressor (Fig. 4a). Surprisingly, RNASE4, a protein of largely unknown function, also inhibited apoptosis in a p53-dependent fashion. We found that like ANG, RNASE4 co-immunoprecipitated with p53 (Fig. 4b), raising the possibility that RNASE4 inhibits p53 through direct protein-protein interaction. The remaining pro-survival SASE genes, *Xdh*, *Abca1*, and *Fth1*, retained their full effects upon p53 depletion (Fig. 4c), indicating that they operate in a p53-independent manner.

**SNCs limit p53 hyperactivity.** The concept suggested by the above observations is that SNCs work to limit p53 activity through multiple mechanisms, which at the surface seems paradoxical because p53, through transcriptional activation of p21, is critical for engaging the senescence program in response to a variety of cellular stressors[52]. However, one possibility would be that once cells become fully senescent, p53 activity needs to be kept in check to suppress its pro-apoptotic functions and ensure SNC survival. We postulated that SASE-mediated induction of *Mdm2*, *Ang*, and *Rnase4* plays a central role in this process. To test this idea, we irradiated MEFs and monitored p53 levels and activity at different stages of senescence evolution (Fig. 5a). Indeed, while p53 and phospho-p53S18 levels rapidly increased after IR and were sustained for several days, they markedly decreased at later time points concurrent with acquisition of the senescence phenotype and accumulation of p16. Importantly, p21 levels remained high in the senescent state despite the marked decrease in p53 activity (Fig. 5a), indicative of uncoupling transcriptional control of p21 by p53. As opposed to proliferating cells or acutely stressed cells, depletion of p53 in SNCs had no impact on p21 levels, providing further evidence for p53-independent expression of p21 in SNCs (Fig. 5b). These data indicate that the level of p53 activity that drives cell cycle arrest in response to genotoxic stress needs to be actively repressed upon entry into senescence in order for SNCs to remain viable. By doing so, SNCs seemingly activate alternative mechanisms to sustain p21 levels that involve SE formation. Although *Mdm2* mRNA levels were

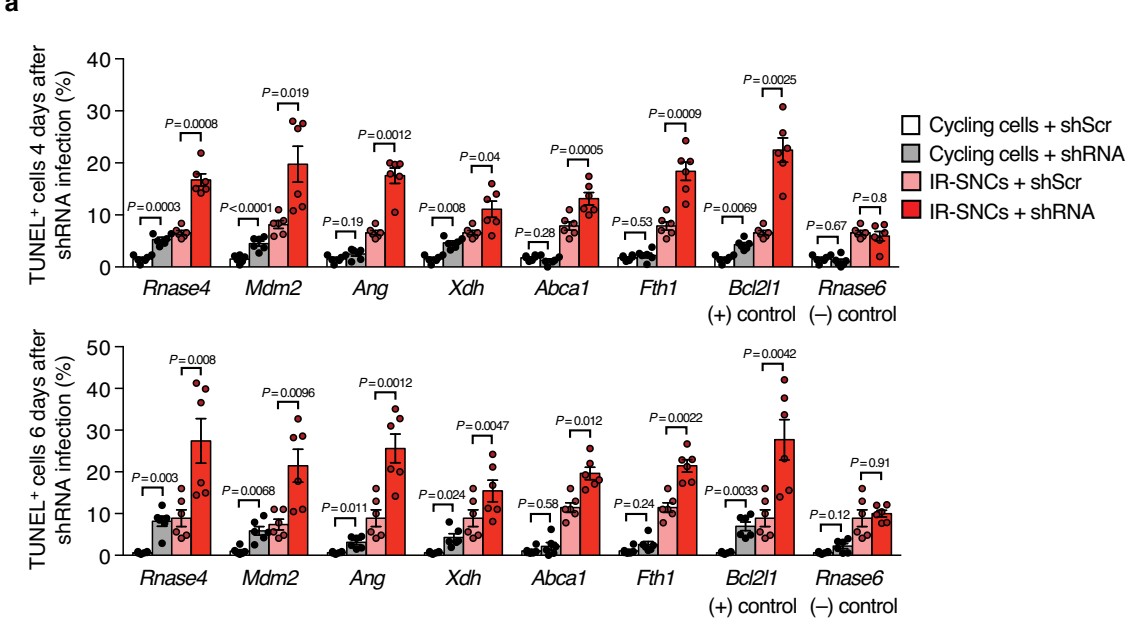

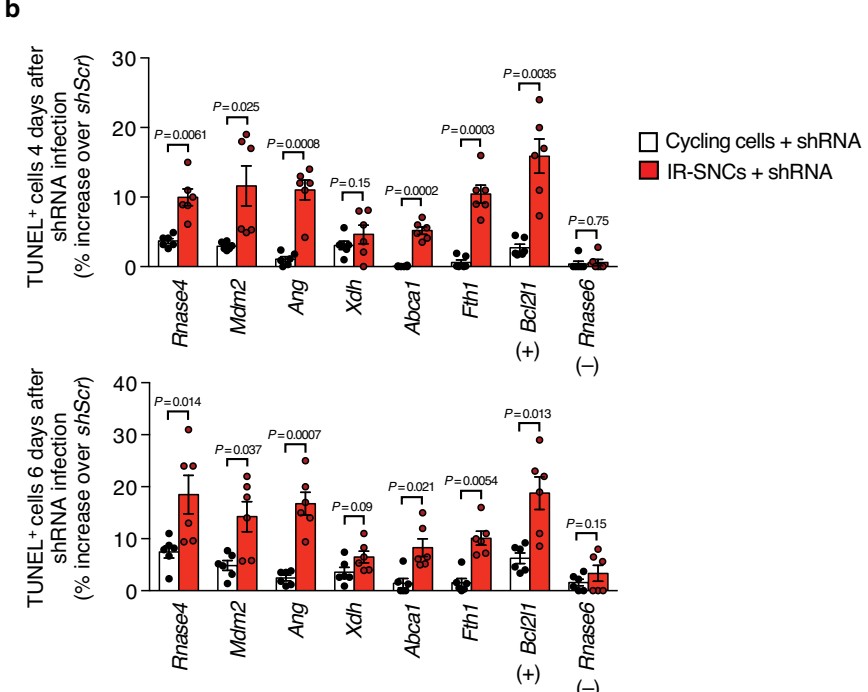

**Fig. 3 SASE genes implicated in SNC survival have limited impact on the viability of cycling cells. a** Quantification of TUNEL-positive IR-senescent or proliferating cells 4 or 6 days after shRNA-mediated SASE gene knockdown. We note that due to the experimental setup some *shScr* control values are used for several comparisons when they were assessed in the same experiment. **b** Increase in TUNEL-positive cells 4 and 6 days after lentiviral infection relative to cells containing scrambled shRNA. Data from **a** were used for this calculation. Data represent means ± SEM. $n = 6$ independent MEF lines pooled from 2 independent experiments in **a**, **b**. Statistics: one-way ANOVA with Sidak's correction in **a**, paired, two-tailed *t*-tests in **b**. Source data are provided as a Source Data file.

elevated in senescent cells regardless of the senescence-inducing stressor (Supplementary Fig. 1c), MDM2 protein levels appeared to decrease (Fig. 5a). Because MDM2 is subject to auto-degradation following DNA damage[53], it is conceivable that increased *Mdm2* transcription by SNCs counteracts mechanisms that downregulate the protein, as such maintaining sufficient MDM2 to keep p53 levels in check for SNC survival.

Next, we depleted *Mdm2* from established SNCs, which we found to increase p53 levels and activity (Fig. 5c). A more modest, but consistent increase in p53 activity was observed with *Ang* knockdown, whereas reduction of *Rnase4* did not induce p53 levels or phosphorylation. Furthermore, knockdown of *Mdm2* or *Ang*, but not *Rnase4 or Bcl2l1*, resulted in transcriptional induction of well-established, pro-apoptotic p53 target genes, including *Puma (Bbc3), Noxa (Pmaip1), Bax* and *Killer/DR5 (Tnfrsf10b)*, prior to apoptosis onset (Fig. 5d). Collectively, these data support the idea that MDM2 and ANG promote SNC viability by inhibiting the transcription of pro-apoptotic p53

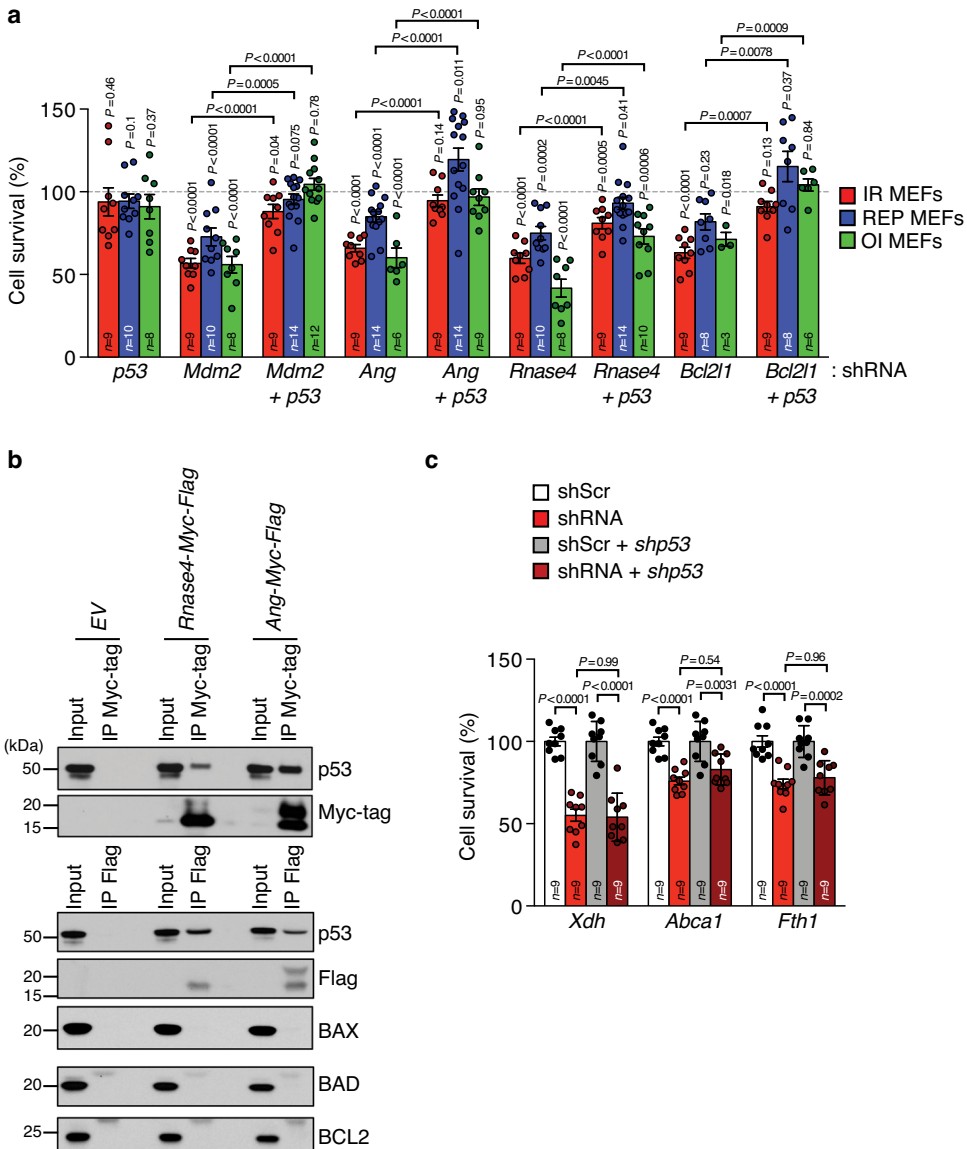

**Fig. 4 Suppression of p53-dependent and -independent pathways contribute to SNC survival. a** Cell survival as measured by MTS assay, of IR-, REP- and OI-SNCs at day 6 after knockdown of the indicated genes. Comparisons were with senescent MEFs containing *Scr*-shRNA or *Scr*- and *p53*-shRNA (100% line). **b** Myc-Flag-tagged RNASE4 or ANG ectopically expressed in HEK-293T cells, precipitated with Myc- or Flag-tag-specific antibodies and analyzed for co-precipitation with the indicated proteins. Western blot result is representative of 2 independent experiments. **c** as in **a** but assessing indicated SASE gene shRNA, *p53* shRNA or both. We note that due to the experimental setup some *shScr* control values are used for several comparisons when they were assessed in the same experiment. Data represent means ± SEM. *n* depict independent MEF lines that were pooled from 2 to 5 independent experiments in **a** or 3 independent experiments in **c**. Statistics: one-way ANOVA with Sidak's correction in **a**, **c**. Source data are provided as a Source Data file.

target genes. On the other hand, RNASE4 appears to act also through p53, but independent of its transcriptional activity.

**ANG and RNASE4 regulate cell fate decisions in response to stress**. The observation that RNASE4 and ANG promote survival of SNCs by limiting p53 activity prompted the question as to whether they might also do so at earlier stages of the DNA damage response when cells are considered pre-senescent. To address this question, we depleted *Rnase4* or *Ang* from MEFs, activated p53 with 4 or 10 Gy ionizing radiation, and measured cell cycle arrest, cell death and cellular senescence 4 days post-irradiation (Fig. 6a). We found that *Rnase4* or *Ang* deficiency decreased cell proliferation at 4 Gy, while increasing apoptosis at 4 and 10 Gy and senescence at 10 Gy (Fig. 6b–d). We complemented these studies with western blot analyses for RNASE4 to determine whether its levels increase in

response to DNA damage. Indeed, RNASE4 was induced as early as 4 days after irradiation and further increased until day 10 post-irradiation when cells were considered senescent (Fig. 6e). Collectively, these data suggest that RNASE4 and ANG are part of the cellular response to DNA damage to attenuate p53-mediated cell fate responses.

**ANG and RNASE4 bind cytoplasmic p53 in response to stress**. To gain insight into the mechanism by which RNASE4 and ANG may regulate p53 and downstream responses, we determined where in the cell RNASE4 and ANG colocalize with p53 at various timepoints after DNA-damage induction. To this end, we stably expressed Myc-Flag-tagged RNASE4 or ANG in MEFs, harvested cells at d0, d4 and d10 after 10 Gy irradiation, and fractionated cells into cytoplasmic (C), membrane/organelle (M),

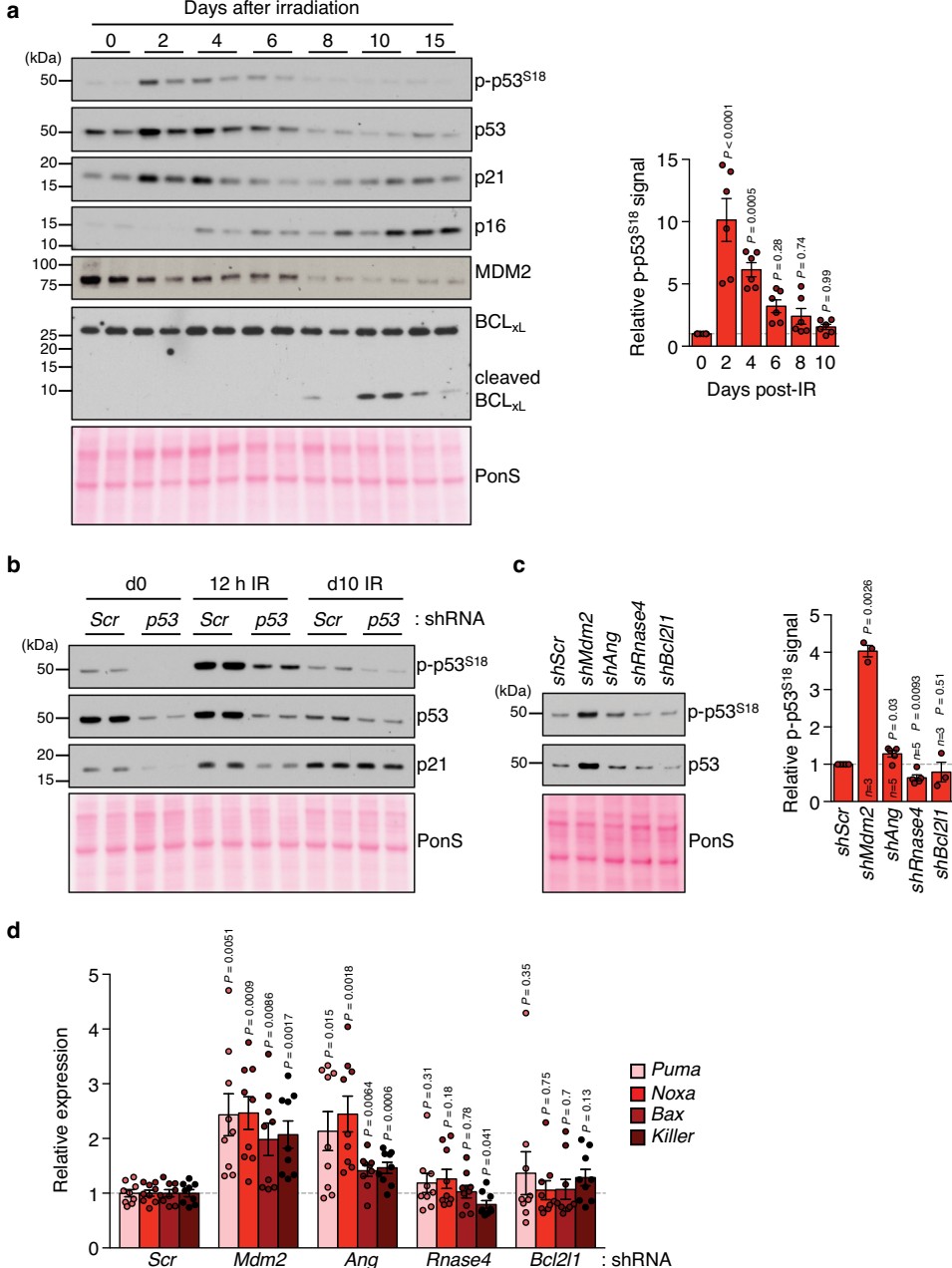

**Fig. 5 SNCs uncouple p21 from p53. a** Western blot analysis of MEF lysates harvested at the indicated time points after irradiation and sequentially probed for the indicated proteins. Note: phosphorylation of mouse p53 serine 18 corresponds to human p53 serine 15. Ponceau S (PonS) staining served as loading control. Densitometric quantification of phospho-p53$^{S18}$ signals normalized to PonS densitometry. **b** Western blot of lysates of the indicated MEF cultures harvested three days after infection with *p53*-shRNA or *Scr*-shRNA. Western blot result is representative of 2 independent experiments. **c** Western blot analysis of IR-senescent MEF lysates harvested 3 days after silencing the indicated SASE genes and densitometric quantification for phospho-p53$^{S18}$ signals as in **a**. **d** RT-qPCR for the indicated p53 target genes 3 days after knockdown in IR-senescent MEFs. Data represent means ± SEM. $n = 6$ independent MEF lines that were pooled from 3 independent experiments in **a**, $n$ indicated in figure pooled from 3 to 5 independent experiments in **c**, $n = 9$ independent MEF lines that were pooled from 3 independent experiments in **d**. Statistics: one-way ANOVA with Sidak's correction in **a**, one-sample $t$-tests in **c**, two-tailed, paired $t$-tests in **d**. Source data are provided as a Source Data file.

and nuclear (N) fractions for analysis by western blotting (Fig. 7a, Supplementary Fig. 5). In the absence or presence of DNA damage, RNASE4 and ANG primarily localized to the soluble and membrane/organelle fractions of the cytoplasm, with relatively small amounts of each protein localizing to the nucleus (Fig. 7b). Accompanying immunofluorescence studies confirmed that RNASE4 and ANG predominantly localize to the cytoplasm, and to a much lesser extent in the nucleus regardless of cell state (Supplementary Fig. 6). RNASE4 and ANG both had granulated

cytoplasmic staining patterns, but areas of high protein concentration unlikely represent mitochondria, as determined by costaining for the mitochondrial marker COX IV (Supplementary Fig. 6). p53 was readily detectable by western blotting in the soluble cytoplasmic and nuclear fractions, but not in the membrane/organelle fractions (Fig. 7b).

The above data together with the observation that RNASE4 and ANG both co-precipitate p53 (Fig. 4b) suggested that RNASE4 and ANG suppress p53 through cytoplasmic

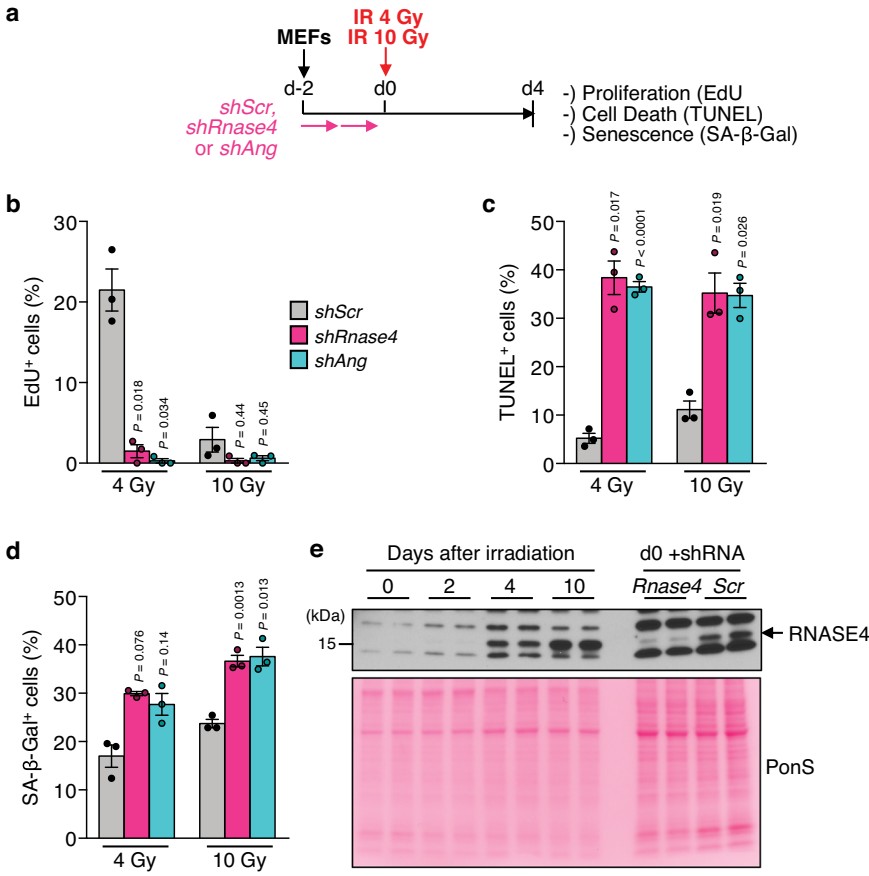

**Fig. 6 RNASE4 and ANG regulate cell fate decisions in response to cellular stress. a** Experimental approach to assess indicated cell fates after *Rnase4* or *Ang* knockdown under stress conditions via 4 Gy or 10 Gy (senescence-inducing amount) of irradiation in MEFs. **b** Proportion of EdU-positive cells. Cells were allowed to incorporate EdU into their DNA for 24 h. **c** TUNEL assay quantification to assess the proportion of dying cells. **d** Quantification of cells positive for senescence-associated β-galactosidase (SA-β-Gal). **e** Western blot analysis of RNASE4 in MEF lysates harvested at the indicated time points after irradiation. Western blot result is representative of 3 independent experiments. *Rnase4* knockdown lysates were assessed in parallel as control. Ponceau S (PonS) staining served as loading control. Data represent means ± SEM. *n* = 3 independent MEF lines from 1 experiment in **b**–**d**. Statistics: one-way ANOVA with Sidak's correction in **b**–**d**. Source data are provided as a Source Data file.

interaction. We examined this possibility by performing proximity ligation assays (PLA) between p53 and Myc-Flag-tagged RNASE4 or ANG on MEFs at d4 and d10 post irradiation. Cytoplasmic PLA signals for p53-RNASE4 and p53-ANG were observed at both timepoints, with signals markedly increasing upon senescence (Fig. 7c, d, and Supplementary Fig. 7a). Myc-Flag-RNASE4 and Myc-Flag-ANG indeed co-precipitated p53 from the soluble cytoplasmic fraction of IR-senescent MEFs (Supplementary Fig. 7b). Collectively, these data demonstrate that RNASE4 and ANG both interact with p53 in the cytoplasm following DNA damage and suggest that RNASE4 and ANG promote senescent and pre-senescent cell survival through a mechanism that involves binding to cytoplasmic p53.

Extended studies showed that RNASE4 or ANG overexpression (OE) had no impact on pre-senescent cell viability after DNA damage (Supplementary Fig. 8a). The same was true for cell cycle arrest and entry into senescence (Supplementary Fig. 8b, c). On the other hand, overexpression of either protein did increase survival once cells had become senescent (Supplementary Fig. 8a). Furthermore, in the absence of DNA damage, RNASE4 or ANG OE had no discernable impact on MEF proliferation, survival or senescence rates (Supplementary Fig. 9a–d). Conversely, cell cycle arrest, apoptosis, and cellular senescence were all markedly increased when *Rnase4* or *Ang* were depleted instead of overexpressed in these experiments (Supplementary Fig. 9e–g).

**Key survival mechanisms of senescent MEFs are conserved in human senescent cells.** The level of conservation of SE-associated genes between mouse and human is limited[54], which also holds true for SASE genes, as we recently reported[9]. However, as there are various other ways to activate gene transcription, conservation at the level of key senescence pathways and mechanisms that promote survival might be substantially higher. Indeed, of the six SASE genes commonly activated in senescent MEFs to promote survival, all but *ANG* were also transcriptionally activated in IR-senescent IMR-90 cells (Fig. 8a). Reminiscent to IR-senescent MEFs, IR-senescent IMR-90 cells also sustained p21 in a p53-independent fashion (Fig. 8b). The high proportion of mouse SASE genes that are also activated in senescent IMR-90 cells, either through nearby human SASEs or via alternative processes, predicted that the here identified mechanisms that promote survival in senescent MEFs are at least partly conserved across species.

To further test this, we knocked-down *MDM2*, *ANG* or *RNASE4* in IR-senescent IMR-90 cells and measured the impact on survival (Fig. 9a, Supplementary Fig. 10a). Indeed, cell viability was significantly reduced with depletion of *MDM2* or *RNASE4*, both of which are activated in senescent IMR-90 cells (Fig. 9a). As a negative control, *ANG* had no impact on survival when reduced, consistent with a lack of activation in senescent IMR-90 cells. As in senescent MEFs, *MDM2* and *RNASE4* stimulated survival by keeping p53 activity in check and co-depletion of *p53* together with

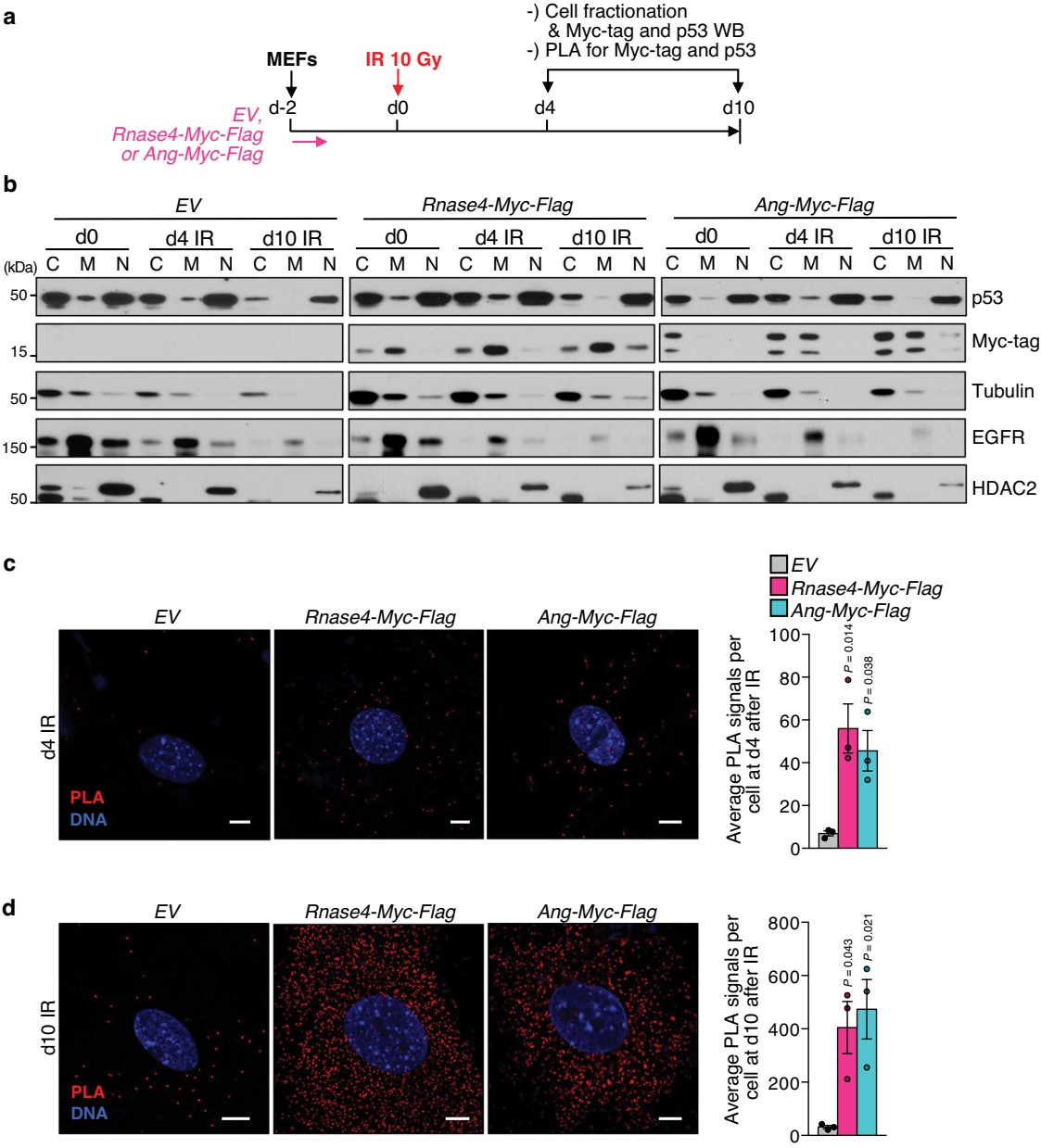

**Fig. 7 RNASE4 and ANG interact with p53 in the cytoplasm in response to stress. a** Experimental set-up to assess interactions between p53 and Myc-Flag-tagged RNASE4 or ANG after senescence-inducing stress. **b** Immunoblots of cell fractions in indicated conditions and timepoints post-irradiation. C, cytoplasmic fraction marked by Tubulin levels; M membrane fraction indicated by EGFR levels; N, nuclear fraction marked by HDAC2 levels. One experiment was performed. **c**, **d** Proximity ligation assay (PLA) on MEFs at indicated timepoints post-irradiation. PLA was assessed between endogenous p53 and Myc-Flag-tagged RNASE4 or ANG. Scale bars, 10 μm. Data represent means ± SEM. $n = 3$ independent MEF lines from 1 experiment in **c**, **d**. Statistics: one-way ANOVA with Sidak's correction in **c**, **d**. Source data are provided as a Source Data file.

*MDM2* or *RNASE4* significantly rescued cell survival (Fig. 9b). Ectopic Myc-Flag-RNASE4 localized to the cytoplasm of pre-senescent (d4 IR) and senescent (d10 IR) IMR-90 cells (Supplementary Fig. 10b). As in MEFs, PLA on IMR-90 cells demonstrated that cytoplasmic Myc-Flag-RNASE4 interacts with p53 both prior to and in the senescent state (Fig. 9c). Again p53-RNASE4 complex formation was more abundant once senescence had occurred. These data suggest that the here reported principles underlying senescent and pre-senescent MEFs survival also apply to human fibroblasts.

## Discussion
High transcriptional heterogeneity observed among SNCs has made it difficult to deepen the molecular mechanistic understanding of

the biological processes and principles that define this intriguing cell state. We hypothesized that combining transcriptional and SE profiling methods would allow for identification of genes that are of particular importance to SNCs. Here, we tested this concept using MEFs and three distinct senescence-inducing stressors. Our functional analyses of genes that were both in the vicinity of a SASE and activated regardless of senescence-inducing stressors provided important insights into the workings of cells in a senescent state. First, SNCs prioritize the activation of genes that contribute to the bioactive secretome or increase survival. Second, while enhanced survival involves both p53-dependent and -independent mechanisms, suppression of p53-mediated apoptosis is a point of emphasis of SNCs and involves both an inhibition of p53's transcription-

**a**

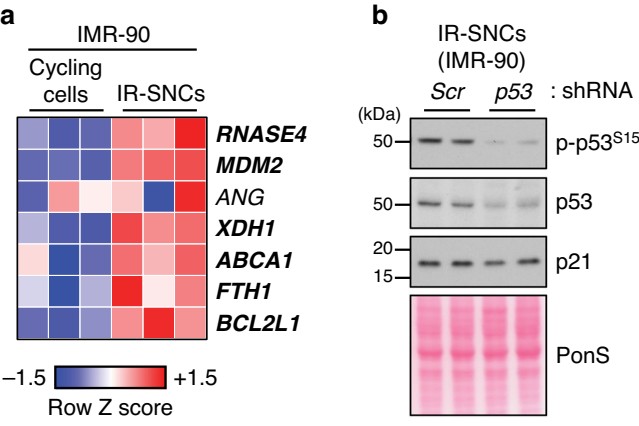

**b**

Fig. 8 RNASE4 expression is elevated in human senescent fibroblasts. a Heatmap of SASE gene expression in IR-senescent IMR-90 cells and proliferating counterparts. Row Z-scores based on RNA-seq data are depicted. Bolded genes are significantly upregulated in IR-SNCs. b Western blot analysis of senescent IMR-90 cells harvested three days after infection with *p53*-shRNA or *Scr*-shRNA. Western blot result is representative of 2 independent experiments. Source data are provided as a Source Data file.

dependent and independent pro-apoptotic actions. Third, SNCs accomplish suppression of p53-mediated apoptosis without compromising p21-dependent cell cycle arrest. Fourth, key discoveries in senescent MEFs seem to largely apply to human IMR-90 cells, thus yielding key entry points for the development of therapeutic approaches aimed at eliminating pathological SNCs.

SASE profiles vary quite dramatically in senescent MEFs driven through different stressors. The number of SASEs in MEFs that are shared with IR-senescent IMR-90 cells is very low, indicating that comparisons of SASEs across stressors, cell types and species is unlikely to be informative. However, we find that by limiting the comparative analysis to one cell type and one species and by introducing only one variable (distinct senescence-inducing stressors), important new general information about SNCs can still be obtained from common SASEs. This is because the genes they target in senescent MEFs are frequently activated through alternative mechanisms in human SNCs. Examples of this are *Rnase4, Abca1, Fth1, and Xdh*, all of which are SASE targets genes in senescent MEFs but not in senescent IMR-90 cells yet still universally upregulated in SNCs of both species.

Activation of p53 and the consequent induction of p21 are early events in response to stress and are important for entry into the senescent state[16]. Therefore, our discovery that the uncoupling of p21 from p53 activity is a critical event for cells in the senescent state was highly unexpected. The most plausible mechanism responsible for p21-p53 uncoupling is the establishment of a conserved SASE near the *Cdkn1a* locus that sustains *p21* expression when p53 levels and activity drop with senescence. Sustaining p21 levels when p53 decreases is imperative because p21 is a unique player in two SNC core properties, cell cycle arrest and expression of secreted factors that constitute the SASP[9]. We find that limiting p53 levels and/or activity in the senescent state is critical in order to stay alive under stress conditions. Earlier studies have demonstrated that SNCs are sensitive to increases in p53 levels, for instance by interfering with p53's ability to bind to inhibitors such as MDM2 and FOXO4[13,55], but the need to actively reduce p53 activity was hitherto not recognized. In fact, we find that SNCs leverage multiple mechanisms to do so, each involving the activation of a distinct SASE gene encoding established or previously unknown inhibitory binding partners of p53 (Supplementary Fig. 11). An earlier study on WI-38 fibroblasts

reported that Nutlin3a, a small molecule that stabilizes p53 by interfering with MDM2-mediated degradation of p53, stimulates p53 transcriptional activity when cells are in a senescent state, but not to the same extent as when they are cycling[56]. As a result, apoptosis rates with Nutlin3a were lower in senescent than in cycling WI-38 cells. These findings contrast our observation that SNCs are much more sensitive to apoptosis upon *MDM2* depletion than cycling cells (Fig. 3b). The same was true for depletion of *RNASE4* or *ANG*, indicating that elevating p53 levels or activity in SNCs consistently induces a robust apoptotic response in senescent MEFs. Complexity also exists concerning p53's role in driving cells into a senescent state[57], with ectopic expression of p53 inhibiting p21-mediated senescence induction in HT1080 fibrosarcoma cells, and instead causing quiescence[58].

In human cancer cells, ANG promotes MDM2-mediated degradation of p53 and inhibits transcription of p53 target genes to prevent apoptosis and increase survival[48,49]. Although consistent with these findings, we find that *Ang* depletion in senescent MEFs results in increased expression of pro-apoptotic p53 target genes, p53 levels remained largely unchanged. Together with our observation that p53 and ANG interact in the cytoplasm of SNCs raises the possibility that ANG may, at least in part, inhibit p53-mediated transcription by limiting its translocation to the nucleus (Supplementary Fig. 11). ANG-p53 complex formation in the cytoplasm may also inhibit apoptosis by preventing p53 from binding to BAX, reminiscent of members of the BCL family. The latter is a plausible mechanism by which RNASE4 inhibits p53-mediated apoptosis, because *Rnase4* depletion from SNCs has no impact on expression of pro-apoptotic target genes or p53 protein levels, and because RNASE4-p53 complex formation in SNCs is cytoplasmic (Supplementary Fig. 11). While RNASE4 induction and complex formation with p53 are abundant in SNCs, we find that both already occur to a lesser extent in earlier stages of the DNA damage response. The same holds true for ANG-p53 complex formation. When RNASE4 or ANG are depleted, cells experiencing DNA damage are highly prone to apoptosis and to a lesser extent senescence, suggesting that both proteins play an integral role in regulating p53-mediated cell fate decisions in the context of the DNA damage response. In extended studies, we found RNASE4 or ANG depletion in the absence of ionizing radiation induces senescence and to a lesser extent to apoptosis, implying that both proteins also regulate cell fate decisions outside the context of DNA damage. One possibility is that cultured MEFs used in these experiments experience p53-activating stressors other than DNA damage. However, p53-independent induction of senescence or apoptosis in the absence of genotoxic stress cannot be excluded, and it will therefore be interesting to further explore the underlying mechanistic details in future studies.

Although the MDM2-p53 interaction is a well-recognized druggable target, its therapeutic applicability has been questioned because widespread untimely or excessive activation of p53 is likely to drive healthy cells into senescence or apoptosis, thereby disrupting tissue integrity[59–63]. Therefore, therapeutics targeting the MDM2-p53 interaction will need to be carefully spatio-temporally tailored to be applicable as SNC-eliminating compounds. Importantly, mice without ANG are viable and do not exhibit any overt phenotype[64], raising the possibility that therapeutic targeting of ANG in the context of senolysis will be more feasible than the direct targeting of MDM2. It should be noted however that the applicability of ANG as a senolytic target may be somewhat limited because its expression was activated in senescent MEFs but not in senescent IMR-90 cells. Both MDM2 and ANG inhibit transcription-dependent apoptosis of p53 by suppressing transcription of pro-apoptotic p53 target genes such as *Bax, Puma* and *Noxa*, which encode proteins that lead to

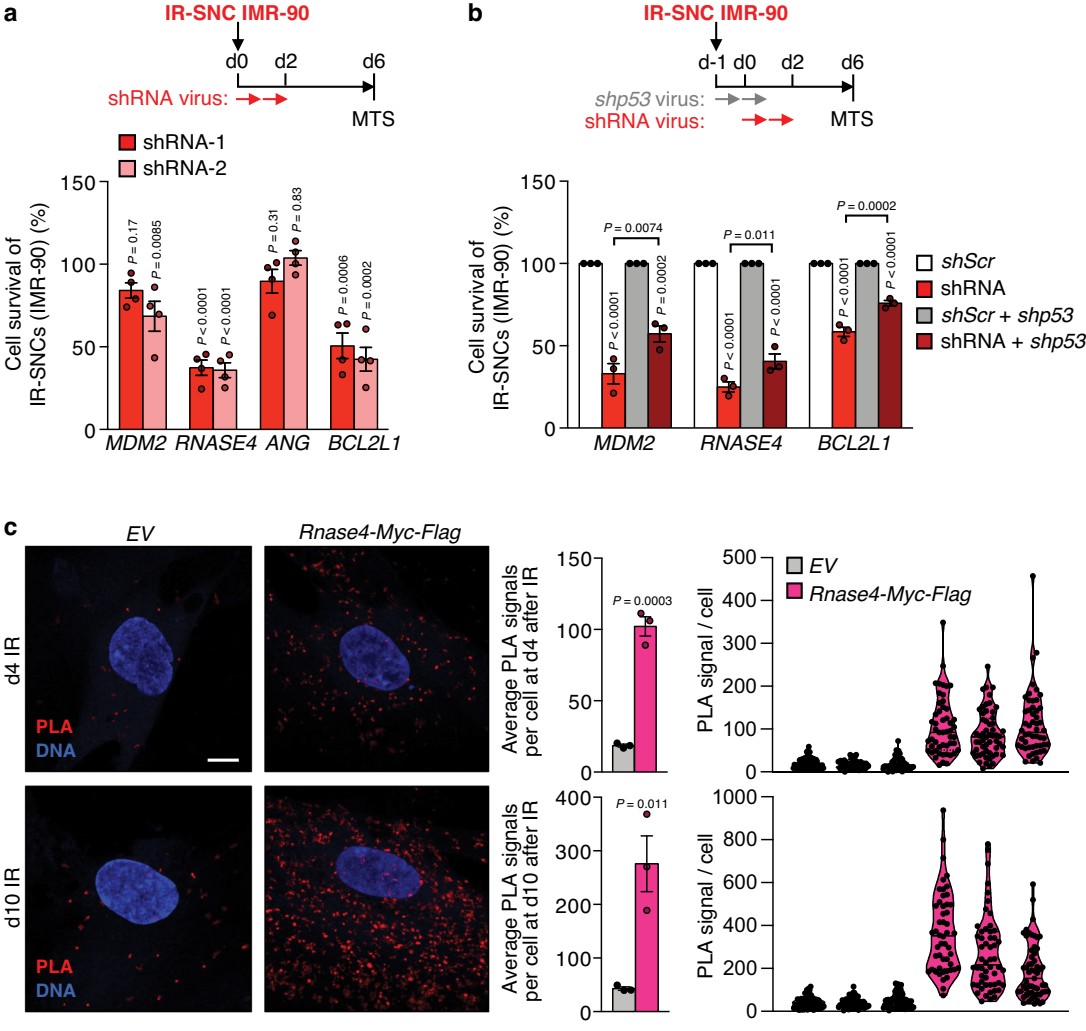

**Fig. 9 Cytoplasmic RNASE4 binds to and inhibits p53 in human senescent fibroblasts. a** Experimental approach and percentage survival of IR-senescent IMR-90 cells 6 days after knockdown of the indicated genes as assessed by MTS assay. Comparisons were made to senescent IMR-90 cells of same cultures infected with *Scr*-shRNA (100%). **b** as in **a** but comparisons were made to senescent IMR-90 cells of same cultures infected with *Scr*-shRNA or *Scr*- and *p53*-shRNA (100%). **c** Representative images and quantification of proximity ligation assay (PLA) to assess spatial interaction of p53 and Myc-Flag-tagged RNASE4 in indicated irradiated IMR-90 cells. Violin blot shows distribution of PLA signals per cell per IMR-90 replicate. One experiment was performed. Scale bars, 10 μm. Data represent means ± SEM. n = 4 **a** or n = 3 **b** independent experiments with each n being an average of 3 technical IMR-90 replicates, n = 3 technical IMR-90 replicates from 1 experiment in **c**. Statistics: one-way ANOVA with Sidak's correction in **a**, **b**, two-tailed, unpaired *t*-tests in **c**. Source data are provided as a Source Data file.

mitochondrial outer membrane permeabilization (MOMP), cytochrome c release and apoptosis[65]. *RNASE4* expression is highly restricted outside the context of cellular stress (per the Human Protein Atlas), suggesting it may be a particularly attractive senotherapeutic target. It is important to note that we find that RNASE4 and ANG have roles during stress response prior to senescence and in the absence of stress, and that prolonged deficiency in either protein can induce senescence in the absence of stress. However, such aspects may be circumvented by minimizing the duration and dose regiment of potential senotherapeutic strategies.

Additionally, we uncovered two SASE genes, *Abca1* and *Fth1*, both of which promote SNC survival independent of p53. ABCA1, an ATP-binding cassette transporters involved in cholesterol efflux, may protect against oxidative stress-induced cell death[66]. Similarly, FTH1 which is involved in iron detoxification and safely stores excess iron as ferritin, could protect against ferroptosis and oxidative stress[67]. These findings further support the idea that SNCs actively combat apoptosis on multiple fronts

and therefore have numerous therapeutically exploitable vulnerabilities for elimination of detrimental SNCs implicated in aging and aging-related diseases.

## Methods

**Cell culture**. Mouse embryonic fibroblasts (MEFs) were generated and cultured as described previously[9]. MEFs were cultured in DMEM (Gibco, #11960) supplemented with 10% heat-inactivated fetal bovine serum, L-glutamine, non-essential amino-acids, sodium pyruvate, gentamicin and β-Mercaptoethanol. MEF lines were expanded at 3% oxygen and used for experiments between passage (P)3 and P6. IMR-90 (ATCC, #CCL-186) were cultured in the same medium as used for MEFs and used for experimentation between P14 and P18. Experimental procedures involving laboratory mice were reviewed and approved by the Institutional Animal Care and Use Committee of the Mayo Clinic.

**Generation of senescent and non-senescent cells**. To generate IR-induced cells, subconfluent cell layers of MEFs or IMR90 cells were exposed to 10 Gy γ-radiation ([137]Caesium source) and cultured for indicated amounts of days with medium changes every 3–4 days. At day 10 post-irradiation, cells were considered senescent based on lack of proliferation, induction of senescence-associated β-Galactosidase, DNA damage marker and a complex senescence-associated secretory phenotype

(SASP)[9]. Generation and enrichment of REP-senescent and OI-senescent MEFs were performed as previously described[9].

**Plasmid constructs**. For shRNA-mediated knockdown experiments, we generated shRNA hairpins cloned into *pLKO.1* vector (Addgene, #10878) as previously described[9]. Briefly, oligo sequences were provided by the RNAi Consortium (TRC, Broad Institute) and). Four to five shRNAs were tested per gene and the two shRNAs with most efficient knockdown potential were used in experiments. A scrambled, non-targeting shRNA (*shScr*; Sigma-Aldrich, #SCH202) was used as a negative control. For shRNA target sequences see Supplementary Table 1. For overexpression studies, we cloned Myc-DDK(Flag)-tagged cDNAs for mouse *Ang* (Origene, #MR200967) and *Rnase4* (Origene, #MR201018) into the lentiviral *pTSIN PGK-puro2* backbone[68,69]. Empty *pTSIN* vector (EV) was used as negative control in overexpression experiments.

**Lentivirus production and cell infection**. Lentiviral particles were generated in HEK-293T cells (ATCC, #CRL-3216) via Lipofectamine 2000 reagent (Invitrogen, #11668) and *pLP1, pLP2, VSV-G* helper plasmids (for *pLKO.1* vectors), or *VSV-G* and *pHR' CMV8.9* (for *pTSIN* vectors). Virus supernatant was harvested 48 h after transfection by filtering HEK-293T supernatants through a 0.45 μm syringe filter. Virus was frozen at −80 °C in small aliquots and freshly thawed for each infection cycle.

**RNA isolation and quantitative PCR**. Cells were seeded in 24-well tissue culture plate at 25,000 cells/well. The next day, medium was replaced with 750 μl virus-containing medium at a ratio of 1:2 (virus supernatant to medium), and infection was repeated the following day. After 48 h of virus exposure, cells were cultured in normal medium for 24 h. Cells were then washed once with PBS and lysed in RLT buffer supplemented with β-Mercaptoethanol according to the RNA extraction protocol. RNA extraction (Qiagen, RNeasy Mini kit, #74104, or RNeasy Micro kit, #74004), cDNA synthesis (Invitrogen, SuperScript III First-Strand Synthesis System, #18080051), and reverse transcription quantitative PCR (RT-qPCR) analysis (Applied Biosystem, SYBR Green Real-Time PCR Master Mix, #4309155) of MEFs and IMR-90 cells were performed according to the manufacturer's instructions avoiding the RNase digestion step. *Tbp* (*TBP* in human) was used as a reference gene for RT-qPCR in mouse and human samples. We note that due to the experimental setup some *shScr* control values may be used for multiple gene knockdown comparisons when they were run in the same experiment. RT-qPCR primer sequences are listed in Supplementary Table 2.

**Western blot, co-immunoprecipitation, and subcellular fractionation**. Immunoprecipitations and western blot analysis were performed as previously described[9,70]. Subcellular fractionation was performed with the Subcellular Protein Fractionation Kit for Cultured Cells (Thermo Scientific, #78840) according to the manufacturer's instructions. Primary antibodies used were: mouse anti-FLAG (Sigma, #F3165; 1:1,000); mouse, anti-Myc-tag (9B11, Cell Signaling, #2276 or #5698; for immunoprecipitation); rabbit, anti-Myc-tag (Cell Signaling, #2272, 1:1,000; for western blot detection); mouse, anti-p53 (Cell Signaling, #2524 S; 1:1,000); mouse, anti-p53-HRP (Santa Cruz, sc-126; 1:1,000 for western blot detection after immunoprecipitations in human samples); rabbit, anti-phospho-p53 S15 (Cell Signaling, #9284; 1:1,000); mouse, anti-p21 (Santa Cruz, sc-53870; 1:8,000); rabbit, anti-p16 (Santa Cruz, sc-1207; 1:1,000), rabbit, anti-BAX (Cell Signaling, #2772; 1:1,000), rabbit, anti-BAD (Cell Signaling, #9292; 1:1000), rabbit, anti-BCL2 (Santa Cruz, sc-492; 1:1,000), mouse anti-BCL-xL (Santa Cruz, sc-8392; 1:1000), rabbit, anti-α-Tubulin (Cell Signaling, #2125; 1:1,000), rabbit, anti-HDAC2 (Abcam, ab7029; 1:1,000) and rabbit, anti-EGFR (Cell Signaling, #71655; 1:1000). The polyclonal rabbit, anti-RNASE4 antibody was generated by GenScript. Recombinant RNASE4 (amino acids 30-129; excluding amino acids 1-29 and 130-148) was expressed in *E. coli* and used to immunize rabbits. Antibody harvest and purification were performed by GenScript. The antibody was validated in-house for use in western blot experiments with various control samples including *Rnase4* knock-down and *RNASE4* overexpression lysates. All antibodies were detected with secondary HRP-conjugated goat, anti-mouse or goat, anti-rabbit antibodies (Jackson Immunoresearch; 1:10,000). Ponceau S staining (0.2% w/v in 5% glacial acetic acid, Sigma-Aldrich, P3504) was used as a loading control. ImageJ was used for densitometric analyses. Phospho-p53$^{S18}$ western blot signals were normalized to total protein content as quantified from Ponceau S staining. Uncropped blots are in source data.

**Cell viability measurements**. Senescent MEFs or IMR-90 cells were seeded in duplicates (IR) or single replicates (REP, OI) in a 96-well flat-bottom tissue culture plate at 2,000 cells/well. The next day, medium was replaced with 120 μl virus-containing medium at a ratio of 1:2 (virus supernatant to medium) and infection was repeated the following day. Twenty-four hours later, the medium was replenished. Six days after the first infection, cells were stained with CellTiter 96 AQueous MTS Reagent (Promega, G1112) with phenazine methosulfate (PMS) (Sigma, P9625) according to manufacturer's instructions. Absorbance was measured at 490 nm using a plate reader (Molecular Devices, SpectraMax M5). For each experiment, knockdown conditions were referred to scrambled shRNA-

infected SNCs of the same culture. For combination infections of *p53*-shRNA and SASE gene-shRNA, *p53* shRNA virus was administrated first (1:2 ratio), 24 h later *p53*-shRNA virus as well as SASE gene-shRNA virus were added (1:1:1 ratio, *shp53* virus to SASE gene-shRNA virus to medium). On the third day, SASE gene-shRNA virus was added (1:2 ratio). MTS assay was performed 6 days after first infection with SASE gene-shRNA virus. Scrambled shRNA-infected SNCs, and scrambled shRNA plus *shp53*-infected SNCs served as controls. We note that due to the experimental setup some *shScr* control values may be used for multiple gene knockdown comparisons when they were run in the same experiment.

**TUNEL assay**. Senescent MEFs were seeded on 10-well chambered slides at 2,000 cells/well and infected with shRNA viruses against SASE genes as described above. Forty-eight hours after medium change, on the fourth and sixth day after the first virus infection, cells were fixed and subjected to TUNEL staining according to the manufacturer's instructions (Sigma-Aldrich, In Situ Cell Death Detection, #11684795910). To assess the proportion of dying cells in non-senescent cell cultures infected with shRNA knockdown viruses or *pTSIN* overexpression viruses, infected cells were seeded on 10-well chambered slides 48 h before the indicted endpoints as described above. At indicated timepoints after knockdown or overexpression, cells were fixed and subjected to TUNEL staining. Cells were counterstained with Hoechst and the percentage of TUNEL-positive cells was determined. At least 200 cells per sample were counted. We note that due to the experimental setup some *shScr* control values may be used for multiple gene knockdown comparisons when they were run in the same experiment.

**EdU incorporation assay**. To assess DNA reduplication after SASE gene knockdown or gene overexpression, MEFs were seeded on 10-well chambered slides at 2,000 cells/well and infected as described above. Forty-eight hours after the first infection, medium was replaced with medium containing 1 μM EdU (5-ethynyl-2′-deoxyuridine, Carbosynth, #NE08701) and cells were allowed to incorporate EdU for 48 h. Four days after the first infection, cells were fixed and subjected to EdU staining according to the manufacturer's instructions (Thermo Scientific, ClickiT Plus EdU Alexa Fluor 488 Imaging Kit, #C10637). To assess the proportion of proliferating non-senescent MEFs after gene knockdown or overexpression, virus-infected cells were seeded as described and allowed to incorporate EdU for 24 h. Cells were counterstained with Hoechst. At least 100 cells per sample were counted to determine the proportion of EdU-positive cells. We note that due to the experimental setup some *shScr* control values may be used for multiple gene knockdown comparisons when they were run in the same experiment.

**Senescence-associated β-Galactosidase assay**. To determine the fraction of senescent cells after gene knockdown or overexpression, virus-infected cells were seeded on 10-well chambered slides (HTC supercured, Thermo Fisher Scientific, #30966 S Black) at 2000 cells/well. On the next day, cells were processed with the Senescence β-Galactosidase Staining Kit (Cell Signaling, #9860) according to the manufacturer's protocol. MEFs were stained for 24 h. Cells were counterstained with Hoechst and at least 100 cells per sample were quantified to assess the proportion of SA-β-Gal-positive cells.

**Immunofluorescence, proximity ligation assay, confocal microscopy**. To assess RNASE4 or ANG localization, cycling cells were first infected with appropriate *pTSIN* viruses once. Cells were then seeded on 10-well chambered slides and, on the next day, either fixed with 4% PFA in PBS for 15 min or irradiated with 10 Gy. At indicated timepoints post-irradiation, cells were fixed with 4% PFA in PBS for 15 min. Cells were permeabilized with ice-cold Acetone for 5 min to preserve mitochondria. Subsequent immunofluorescence and confocal microscopy were performed as previously described[9]. Primary antibodies used were: mouse, anti-COX IV (Abcam, ab33985; 1:100) and rabbit anti-Myc-tag (Cell Signaling, #2272, 1:100). Secondary antibodies were: goat, anti-mouse Alexa Fluor 488 (Invitrogen, #A11029; 1:250) and goat, anti-rabbit Alexa Fluor 594 (Invitrogen, #A11012; 1:250). Image acquisition was performed on a confocal laser-scanning microscope (LSM 880; Zeiss) on an Axio Observer Z1 inverted microscope with spectral detectors (32ch 2PMT GaAsP; Zeiss) and a water immersion lens (C-Apochromate 40X/1.2 NA Korr. FCS; Zeiss). To assess spatial protein-protein interactions between p53 and RNASE4 or ANG using proximity ligation, cells were prepared as described above. Proximity ligation assay was performed using the Duolink In Situ Orange Starter Kit Mouse/Rabbit (Sigma-Aldrich, #DUO92102) according to the manufacturer's instructions. Antibodies used for experiments in both MEFs and IMR-90 cells were mouse, anti-p53 (Cell Signaling, #2524 S; 1:50) and rabbit, anti-Myc-tag (Cell Signaling, #2272, 1:100). The Duolink PLA Control Kit—PPI (Sigma-Aldrich, #DUO92202) was employed in parallel as additional technical quality control. PLA signals were imaged on the same confocal laser-scanning microscope as described above. PLA signals per cell were quantified via ImageJ using the particle analysis tool. At least 50 cells per sample were quantified.

**Statistical analysis**. GraphPad Prism software was used for statistical analyses. Unless otherwise stated, student's two-tailed, paired *t*-tests (in MEFs) or student's two-tailed, unpaired *t*-tests (in IMR-90 cells) were used for pairwise comparisons

involving two groups. For experiments involving three or more groups, one-way analysis of variance (ANOVA) with Sidak's correction was performed.

**Reporting summary**. Further information on research design is available in the Nature Research Reporting Summary linked to this article.

## Data availability

Source data are provided with this paper. ChIP-seq and RNA-seq data sets have been previously deposited in the Gene Expression Omnibus database under accession number GSE117278. Details on ChIP-seq and RNA-seq sample generation and analyses were reported previously[9]. Two to three independent MEF lines or three technical IMR-90 cell replicates were used for ChIP-seq experiments and three independent MEF lines or three technical IMR-90 cell replicates were used for RNA-seq experiments. Source data are provided with this paper.

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

## Acknowledgements

We thank Jeremy T. Stutchman, Ismail Can, Erik-Jan van Deursen, Naomi Hamada, Jan Grasic, Jazeel F. Limzerwala, and Do Young Lim for their help with functional studies, as well as Christian A. Ross, Tamas Ordog, and Jeong-Heon Lee for assistance with the acquisition and interpretation of SE data. We are grateful to Remi-Martin Laberge for advice on survival studies and Marten H. Hofker and Bart van de Sluis for intellectual input. This work was supported by grants from Mayo Clinic's Center for Biomedical Discovery, the Paul F. Glenn Foundation, the Keck Foundation and the US National Institutes of Health (NIH) grants R01 AG057493, R01 CA096985, and R01 AG56318.

## Author contributions

J.M.v.D. led the study. J.M.v.D., H.L., and I.S. conceptualized the study. I.S., C.Z., and H.L. identified super-enhancers. I.S. and J.M.v.D. designed functional experiments and I.S. performed most of these with the assistance of C.C.S. K.B.J. performed functional studies on ANG and RNASE4 with the assistance of R.O.V.F. D.J.B. provided intellectual input and support throughout the study. I.S. and J.M.v.D. wrote the manuscript. All authors contributed to data acquisition, analysis and/or interpretation, and edited the manuscript.

## Competing interests

J.M.v.D. is a co-founder of and holds equity in Unity Biotechnology (a company developing senolytic medicines), and Cavalry Biosciences. J.M.v.D. and D.J.B. are inventors on patents licensed to or filed by Unity Biotechnology. All other authors declare no competing interest. This research has been reviewed by the Mayo Clinic Conflict of Interest Review Board and is being conducted in compliance with Mayo Clinic conflict of interest policies.
