## [Peer Review File · Nature Communications]

Senescent cells limit P53 activity via multiple mechanisms to remain viableREVIEWER COMMENTS

Reviewer #1 (Remarks to the Author):

In this manuscript, Sturmlechner and colleagues have systematically examined the mRNAs and proteins encoded by genes transcribed from senescence-associated super-enhancers (SASEs), with a focus on their possible function in the major traits of senescent cells. They found that the products of SASE genes are implicated in the senescence-associated survival phenotype by suppressing P53-mediated apoptosis.

Conceptually, this study is a nice example of senescence drivers (p53 in this case) triggering a feedback loop that reduces a potentially harmful trait (p53-driven apoptosis). This model is supported by the various silencing interventions, which revert the protective status of senescent cells and promote apoptosis. Unfortunately, while the assembly of observations allow the authors to propose a novel global program for p53, the individual observations are not themselves novel. The study is also quite descriptive at this stage, although the primary concern is with novelty.

1) I am not sure I fully agree with the authors' statement that their discoveries argue "against the established view that P53 activation is a key driver of the senescence program." I believe the field of senescence has advanced substantially, as reviewed recently for example by Giordano and colleagues (PMID: 32182711), but proposed earlier (e.g., PMID: 20457898). The notion that a key aspect of the senescent program is the inhibition of the apoptotic function of p53 is not as novel as the authors propose. For example, the regulation of p53 by MDM2 in senescence is extensively studied, but the transcriptional regulation of RNASE4 and ACE expression by p53 was reported (PMID: 24659782), as has been the interaction of RNASE4 with p53 has been reported (<https://thebiogrid.org/111967/summary/homo-sapiens/rnase4.html>) and other individual aspects of the authors' model.

2) If the p53-regulated proteins MCM2, RNASE4, and ANG (and BCL2L1) are important for repressing the proapoptotic actions of p53, what is the kinetics of expression of these proteins in senescence? It would be expected to occur before p53 levels declined. Again, here, the novelty of these observations is limited, given that all 3 proteins have been shown to promote survival in different scenarios. It is interesting that they appear coordinately expressed in the paradigms shown here (IR, replicative senescence, etc), but again, novelty appears limited.

3) It is a little confusing that the authors begin the study by talking about super-enhancers, but the manuscript focuses on mRNAs differentially expressed in different senescence models, regardless of how they came to be elevated, something that other labs have also done. Presumably the authors have addressed these points on super-enhancer-driven transcription in the manuscript listed as reference 42, which did not appear to be provided in the submitted materials.

Reviewer #2 (Remarks to the Author):

Sturmlechner et al. present a manuscript in which they identify a panel of candidate senescence-associated super-enhancer genes (SASEs) following H3K27Ac-chromatin immunoprecipitation followed by sequencing (ChIPseq). To identify the SASEs, the authors employ the following models of MEF senescence: high-dose g-irradiation (IR), extensive replication (REP) or oncogenic signaling by KRASG12V (OI), and in the later stages of the

paper human IR-senescent IMR-90 cells. The papers primarily focuses on RNase4 and ANG. shRNA mediated knockdown of these proposed SASEs does not enable resumption of the cell cycle in senescent cells but results in reduced cell survival and increased TUNEL +ve cells in senescence versus control cells. shRNA KD of RNase4 (but not ANG) results in the significant reduction in at least 1 out of 4 SASP genes tested. The reduced cell survival following KD of RNase4 or ANG can be rescued to some extent by concomitant p53 KD. RNase4 and ANG are shown to co-IP with p53.

The authors then explore the temporal kinetics of p53 (and p53S18) protein expression following irradiation, and show a convincing initial increase followed by a gradual decline in both up to a 15 day time point, and present data to uncouple p21 protein levels from p53. KD of ANG results in a significant increase of pro-apoptotic p53 targets (Puma, Noxa, Bax, Killer) and p21 (the later is confusing), whilst RNASE4 KD significantly decreases Killer (and is thus proposed to act via p53, independent of its transcriptional activity).

RNASE4 is shown to be a transcriptionally upregulated in IR-IMR-90s, whilst ANG is not. KD of RNASE4 in IR-IMR90s reduced cell survival, and similar temporal kinetics for p53 following senescence onset are demonstrated.

The manuscript is a very clearly written, with a solid logic flow, generally sound statements, although some of the conclusions could be tempered slightly. The presentation of the SASEs is novel and interesting. However, my overall impression of the manuscript is that the findings are not fully explored to warrant publication in a journal of this caliber. Detailed below are some suggestions that would help improve the manuscript further.

Suggested revisions

- 1. Whilst RNaseq data indicates that expression of RNASE4 and ANG is upregulated in multiple models of MEF senescence, no evidence is provided that this results in an increase in protein expression. It will be important for the author to show that this occurs as part of their model. What happens to the levels of RNASE4 and ANG over a similar timecourse to that presented for p53?**
- 2. The authors could additionally explore whether overexpression of RNase4 and ANG is able to induce senescence, or not, using existing constructs. The manuscript would be further strengthened by experiments to explore whether p53 co-localises with either of these proteins in the early stages of senescence (e.g. PLA).**
- 3. Can the authors provide evidence that RNASE4 is also a SASE in the IMR90s? At present only expression data is provided.**
- 4. The paper diverges when the authors suggest that ANG promotes viability by inhibiting the transcription of pro-apoptotic p53 target genes, whilst RNASE4 acts independently of p53 transcription activity. The authors should consider exploring one of the angles in further details to provide more insight into the role of these SASEs in restraining senescent cell apoptosis.**
- 5. Are RNASE4 and ANG drivers of senescence (i.e. essential for restraining an apoptotic fate) or are the passengers in the establishment of the phenotype that function to restrain apoptosis following senescence establishment?**
- 6. As currently presented, the authors do not explore the SE feature at all in this manuscript. For example, what would mutation of the candidate SE mean for the establishment and maintenance of senescence?**
- 7. Of note, p53 has previously been linked to RNASE4 (PMID 23284306) and ANG (PMID 22266868) – the authors should explore this publication in their discussion.**
- 8. It is interesting that both RNASE4 and ANG (RNASE5) are members of the RNase A superfamily. What happens to the other eight canonical RNases in IR-IMR90s? For example, is another family member able to act in a similar fashion to murine ANG? Could this provide a route into a more complete picture?**

Reviewer #3 (Remarks to the Author):

In this manuscript the authors investigate and present the important role of SASE target genes in fundamental properties of senescent cells. They find that a large number of SASE target genes that are activated upon induction of senescence, regardless of the inducer, promote the expression of SASP proteins and regulate the survival of senescent cells by suppressing p53-mediated apoptosis. Their findings are very interesting as they shed light on the complicated landscape of senescence.

Overall, this is a very good and promising piece of work that could be accepted for publication as long as the authors address some important issues mentioned below.

Major points

- 1. Experimental validation of the senescence phenotype of IR-, REP-, and OI-senescent MEFs should be provided.**
- 2. In figure 2a and in supplementary figure 4a, the images are too bright.**
- 3. In supplementary figure 2 it is important to show the value of shScr control.**
- 4. Western blotting would be better supported by an actual loading control marker such as GAPDH, actin etc. rather than Ponceau S staining.**

Minor points

- 1. In the following part of the manuscript text: " These data are the first to indicate that the level of P53 activity that drives cell-cell arrest in response to genotoxic stress needs to be actively repressed upon entry into senescence in order to for SNCs to remain viable. By doing so, SNCs seemingly activate alternative mechanisms to sustain P21 levels that involve SE formation" please replace "cell-cell arrest" with "cell cycle arrest".**

Reviewer #4 (Remarks to the Author):

This is a very interesting manuscript with potentially impactful results with respect to the understanding of epigenetic-molecular mechanisms driving senescence. The entire manuscript relies on the identification of SASEs and SASE-associated genes, and thus the analysis that is shown in supplemental Fig. S1 is of paramount importance to the rest of this body of work. There are concerns with some of the methodology, including the identification of SEs and assigned genes that should be addressed below:

- 1. Supplemental Fig. 1a. Controls appear appropriate to eliminate genes responsive to non-senesence-induced stimuli.**
- 2. Materials and Methods, p19: How many biologic replicates were performed per condition for ChIP-seq? It is standard to perform at least two replicates. If that was not done, at least another replicate should be performed.**
- 3. Materials and Methods, p20 and Suppl. Fig. 1b: regions defining SEs were excluded if they were within 2kb of any TSS, and yet in Suppl. Fig. 1b. every SASE is <2kb from the TSS of the three genes shown, and in fact covers the gene body or TSS of every gene shown. Please explain then why these are considered SASEs.**
- 4. p20, identification of SEs using methodology of Whyte et al., 2013 (Cell): Unfortunately, H3K27ac ChIP-seq alone cannot be used to identify a super-enhancer. Whyte et al., had determined that only 21% (155/725) "SE"s identified by this method (occupancy slope ≥ 1) by H3K27Ac ChIP, were identified as SEs by MED1 or transcription factor ChIP. In order for the authors to properly classify the 50 SASEs, they will need to confirm by MED1 ChIP.**
- 5. suppl fig. 1a. and p21, assignment of SE-associated genes. 50kb seems to be an arbitrary distance to assign SASE-genes given that SEs can affect genes anywhere within a topological domain, which can be as large as 1Mb. Could the authors describe where they came up with +/-50kb distance to assign genes to super-enhancers? In the paper they cite (Whyte et al., 2013, Cell) enhancers and SEs were assigned to nearest gene, regardless of**

distance. They also found that 93% of SEs were found within the same topological domain, so this approach can be added to confirm gene assignments.

6. Materials and Methods, p22: How many biologic replicates were performed per condition for RNA-seq? It is standard to perform at least two replicates. If that was not done, at least another replicate should be performed.

RESPONSE TO THE COMMENTS OF THE REVIEWERS

We thank the reviewers for evaluating our study and providing us with helpful comments and suggestions for improving our manuscript based on their feedback. We greatly appreciate their time and efforts and are happy that the suggested experiments were very insightful and instrumental in further advancing the molecular-mechanistic depth and novelty of our manuscript.

Guided by the specific experimental suggestions of reviewer 2 and the more general request for increased novelty of reviewer 1, we fleshed out the mechanisms by which ANG and RNASE4 inhibit p53 to promote senescent cell survival. Furthermore, we explored the extent to which these mechanisms might apply to p53 regulation in the context of the DNA damage response before major cell fate decisions such as apoptosis and cellular senescence are made. Indeed, these are logical extensions, particularly because the ANG-p53 functional relationship has merely been studied in the context of cancer cell survival, not in the context of DNA damage or senescence, and because no information has been published about the RNASE4-p53 functional relationship in any context. Briefly, the main new findings of these extended studies are:

- 1) ANG- or RNASE4-depleted MEFs are prone to apoptosis and cellular senescence when exposed to ionizing radiation, indicating that both proteins act to inhibit p53 activity not only when cells have become senescent, but also when non-senescent cells respond to DNA damage. These experiments suggest that ANG and RNASE4 are an integral part to the cellular response to DNA damage. These experiments are presented in Fig. 6 and described on pages 8 (lines 22-26) and 9 (lines 1-8).
- 2) That ANG-p53 and RNASE4-p53 complex formation in MEFs occurs in the cytoplasmic compartment in response to DNA damage, and this further intensifies when cells enter a state of senescence. These experiments suggest that, both in presenescent and senescent cells, ANG and RNASE4 exert their inhibitory effects on p53 (at least in part) through direct interaction, reminiscent of BCL2 and BCL-xL. These experiments are presented in Fig. 7 and Supplementary Figs. 5, 6 and 7, and described on pages 9 (lines 10-25) and 10 (lines 1-11).
- 3) RNASE4, which we showed is also a SASE gene in human cells, also engages in complex formation with p53 in the cytoplasm both in response to DNA damage, and more robustly once cells have entered a state of senescence. These data provide evidence of functional conservation in RNASE4 function between mice and humans. These experiments are presented in Fig. 9 and Supplementary Fig. 10 and described on page 11 (lines 14-18).
- 4) ANG- or RNASE4-depletion markedly increases the rate of senescence in the absence of DNA damage, but their ectopic overexpression *per se* has no impact. These experiments are presented in Supplementary Fig. 9 and described on page 10 (lines 17-20).

Collectively, our extended experimentation resulted in three new main figures (Figs. 6, 7 and 9) and six new supplementary figures (Figs. 5, 6, 7, 8, 9 and 10). The raw data that went into the main and supplementary figures have been added to the source file.

We would like to point out that in the original submission we referred to an unpublished manuscript that also used a senescence-associated super-enhancer-based approach. This manuscript, which reports the role of p21 in immunosurveillance of stressed cells, has since been published in *Science* and includes detailed information about our approach to identifying senescence-associated super-enhancers (Sturmlechner *et al.*, 2021: PMID 34709885). In our revised manuscript, we are referring to the *Science* paper at various places, including the Introduction (page 4, lines 11-17, and), the Results (page 5 lines 5-6, and page 10 lines 23-24), the discussion

(page 13 lines 4-6) and the Methods section (page 15 line 20, page 16 lines 7-10, 13-14, page 21 lines 2-3, and 22 lines 3-4).

Below we provide detailed responses to each of the reviewers' comments. The Reviewer comments and our responses are in black and blue font, respectively.

POINT-BY-POINT RESPONSE TO THE REVIEWERS' COMMENTS

Reviewer #1:

In this manuscript, Sturmlechner and colleagues have systematically examined the mRNAs and proteins encoded by genes transcribed from senescence-associated super-enhancers (SASEs), with a focus on their possible function in the major traits of senescent cells. They found that the products of SASE genes are implicated in the senescence-associated survival phenotype by suppressing P53-mediated apoptosis.

Conceptually, this study is a nice example of senescence drivers (p53 in this case) triggering a feedback loop that reduces a potentially harmful trait (p53-driven apoptosis). This model is supported by the various silencing interventions, which revert the protective status of senescent cells and promote apoptosis. Unfortunately, while the assembly of observations allow the authors to propose a novel global program for p53, the individual observations are not themselves novel. The study is also quite descriptive at this stage, although the primary concern is with novelty.

We are very grateful for the reviewer's incisive comments and suggestions which challenged us to not only better articulate the novelty and importance of our study, but also to implement new experiments that further advanced our study in this regard. Please see below for details.

- 1) I am not sure I fully agree with the authors' statement that their discoveries argue "against the established view that P53 activation is a key driver of the senescence program." I believe the field of senescence has advanced substantially, as reviewed recently for example by Giordano and colleagues (PMID: 32182711), but proposed earlier (e.g., PMID: 20457898).

Response:

We had missed the study by Demidenko *et al.* and thank the reviewer for pointing it out. Indeed, while most of the existing literature describes p53 as a key player in the establishment of cellular senescence, the report of Demidenko *et al.* cited by the reviewer challenges this concept. The study comprehensively shows that ectopic expression of p21 in HT1080 fibrosarcoma cells induces senescence and that coexpression of p53 prevents activation of the senescence program and instead induces quiescence. With these findings, the authors demonstrate that p53 can act as a suppressor of senescence and that it therefore not always acts as a driver of senescence. This dual effect of p53 of promoting or inhibiting the senescence program is also discussed by Giordano and colleagues in a recent review article mentioned by the referee.

Where Demidenko and colleagues conclude that p53 activity prevents entry into senescence, our study focuses on cells that are in a state of senescence and report that such cells limit

p53 activity through senescence-associated super-enhancers, which was previously unknown or not discussed. Furthermore, guided by the reviewer's feedback, we have worked to further dissect the mechanisms by which super-enhancer-associated genes keep p53 in check during senescence, and where these mechanisms were novel, we have explored their relevance in the context of the DNA damage response.

That said, we wholeheartedly agree with the reviewer that our statement is confusing, and we have therefore adapted it. Please see the results section on page 7, lines 14-20, and the discussion on page 13, lines 19-21. Furthermore, we have incorporated the review article by Giordano and colleagues and the paper by Demidenko *et al.* that were pointed out by the reviewer into the discussion section of the manuscript (page 13, lines 19-21).

The notion that a key aspect of the senescent program is the inhibition of the apoptotic function of p53 is not as novel as the authors propose. For example, the regulation of p53 by MDM2 in senescence is extensively studied, but the transcriptional regulation of RNASE4 and ACE expression by p53 was reported (PMID: 24659782), as has been the interaction of RNASE4 with p53 has been reported (<https://thebiogrid.org/111967/summary/homo-sapiens/rnase4.html>) and other individual aspects of the authors' model.

Response:

We agree that inhibition of the pro-apoptotic function of p53 is a key aspect of the senescent program *per se* is not novel. Interventions that increase p53 levels in senescent cells, such as inhibiting MDM2 or FOXO4, have illustrated this, as was highlighted on page 11 (lines 4-7) of the original manuscript. So, it wasn't our intention to claim that inhibition of the pro-apoptotic functions of p53 *per se* is novel.

However, what to our knowledge has not been previously reported is that, after cells become senescent, they downregulate p53 levels and engage multiple mechanisms to inhibit its pro-apoptotic functions, some of which are entirely novel as further discussed below. We believe the discovery that senescent cells leverage multiple mechanisms to control p53 is also important in that it offers novel entry points for the development of targeted experimental therapeutics that interfere with these mechanisms to eliminate pathological senescent cells. With this therapeutic angle in mind, our focus for this study initially was specifically on cells that had entered a senescent state.

Regarding MDM2, we agree that it has been extensively reported that MDM2 inhibition promotes senescence or a senescent-like state (Efeyan *et al.* 2007 PMID 17671205, Shen *et al.* 2010 PMID 20489208, Villalonga-Planells *et al.* 2011 PMID 21483692, Manfe *et al.* 2012 PMID 22377766). However, to our knowledge, the only study on MDM2 inhibition in the context of cells in a senescent state is an osteoarthritis study by the Elisseeff lab in which a small molecule MDM2 antagonist was used as a senolytic agent (Jeon *et al.* 2017 PMID 28436958) This study is referenced in the discussion of our manuscript.

In cancer cell lines, ANG binds p53 to promote MDM2-mediated degradation of p53 and inhibit transcription of p53 target genes, including pro-apoptotic target genes, thereby increasing cancer cell survival. All this has been reported in the paper mentioned by the reviewer, Sadagopan *et al.* 2012 (PMID 22266868). We had included a reference to this paper, and it was not our intention to claim that ANG is a novel regulator of p53 *per se*. However, a p53-dependent role for

ANG in senescent cells or the senescence program at large has not been reported. Furthermore, we have now included extended studies in non-transformed cells addressing the role of ANG in early fate decisions (prior to entry into senescence) in response to DNA damage, providing important new insights into its biological relevance and mechanism of action.

Concerning RNASE4, there is to our knowledge no literature on its role as a regulator of p53. The study cited by the reviewer (Sheng *et al.*, 2014, PMID 24659782) reports that p53 is one of 63 transcription factors that could regulate *ANG* and *RNASE4* gene promoter activity. Furthermore, the BioGrid entry mentioned involves an shRNA-based genetic screen by Xie *et al.* (2013, PMID 23284306) to identify genes that are preferentially required for proliferation of p53-deficient HCT116 cells relative to p53-sufficient HCT116 cells. *RNASE4* was identified as one of 103 such genes in the screen. How *RNASE4* depletion promotes proliferation of p53-deficient cells and the nature of the RNASE4-p53 relationship were not addressed.

Furthermore, as for ANG, we have extended our molecular-mechanistic work on RNASE4 in the context of senescence and cell fate decisions to prior to senescence, which further adds to the overall novelty and importance of our study. Also, we have modified the discussion to better articulate the novelty of our initial findings and to include the added novelty of our extended studies. See revised manuscript page 13 (lines 23-26) and page 14 (lines 1-13).

- 2) If the p53-regulated proteins MCM2, RNASE4, and ANG (and BCL2L1) are important for repressing the proapoptotic actions of p53, what is the kinetics of expression of these proteins in senescence? It would be expected to occur before p53 levels declined. Again, here, the novelty of these observations is limited, given that all 3 proteins have been shown to promote survival in different scenarios. It is interesting that they appear coordinately expressed in the paradigms shown here (IR, replicative senescence, etc), but again, novelty appears limited.

Response:

As requested, we examined whether increased expression of MDM2, RNASE4 and ANG occurred before the observed decline in p53 levels. The reviewer's comment nicely dovetailed with reviewer 2's request for studies into the role of ANG and RNASE4 in regulating p53 activity prior to senescence entry. These extended studies indicate that both ANG and RNASE4 are an integral part of the machinery that regulates p53 activity following DNA damage and determines cell fate, in addition to controlling cell fate (promoting survival) once cells have become senescent.

Regarding MDM2, although MDM2 mRNA levels are elevated in the senescent state (10 days post-irradiation), the corresponding protein levels are not increased and even appear to decrease as cells become senescent. Please see the newly added western blot in Fig. 5a of the revised manuscript. At first instance, this result was somewhat puzzling, but MDM2 is known to auto-degrade quickly after DNA damage (Stommel *et al.* 2004, PMID 15029243). It is therefore conceivable that increased *Mdm2* transcription is necessary to counteract mechanisms that downregulate MDM2 protein, as such maintaining sufficient MDM2 to keep p53 levels in check for senescent cell survival. See also text on page 8 (lines 5-10).

For RNASE4 we encountered technical difficulties, in that after critically testing commercially available antibodies, we were unable to obtain antibodies that detect RNASE4 on western

blots. To bypass this problem, we generated rabbit polyclonal antibodies against murine RNASE4. As shown in Fig. 6e of the revised manuscript, these antibodies detect RNASE4 on western blots of MEF lysates (validated with shRNA-mediated *Rnase4* knockdown in MEFs). In using this antibody in studies designed to explore the role of RNASE4 in p53 regulation in the broader context of the cellular response to DNA damage, we found that RNASE4 levels are elevated as early as 4 days post-irradiation and further increase by day 10 post-irradiation when most cells have entered a senescent state (Fig. 6e of the revised manuscript). These results indicate that, as the reviewer alluded to, the formation of a SASE further boosts RNASE4 in the senescent state. We note that the initial increase in RNASE4 expression following DNA damage is p21 and Rb1 dependent and part of the p21-Associated Secretory Phenotype (PASP). Please see Supplementary Data Table 6 of Sturmlechner *et al.*, 2021 (PMID 34709885).

Concerning ANG, our attempts to identify suitable antibodies for western blotting were unsuccessful. Therefore, we were unable to evaluate ANG protein levels by western blot analysis at various stages after radiation. However, we did obtain valuable information about the kinetics of the p53-ANG (and p53-RNASE4) interaction by performing proximity ligation assays (PLA) on MEFs stably expressing Myc-Flag-ANG (or Myc-Flag-RNASE4) after DNA damage prior to senescence, and after entry into senescence. We confirmed that consistent with the published data on endogenous ANG (for instance Kishimoto *et al.* 2005, PMID 15558023), ectopically expressed Myc-Flag-ANG localizes primarily to the cytoplasmic compartment (showing a granular texture). The results of these experiments are shown in Fig. 7c-d and Supplementary Fig. 6. They show that both ANG and RNASE4 indeed interact with p53 in the cytoplasm in a DNA damage-dependent fashion. Interaction already occurs prior to senescence, but the abundance of p53-RNASE4 and p53-ANG complex formation markedly increased in the senescent state. In accompanying experiments involving shRNA depletion of *Rnase4* or *Ang*, we show that these two proteins are an integral part of cell fate decisions in both the earlier stages of the DNA damage response and in the senescent state (these data are presented in Fig. 6). Please see also point 1 of reviewer 2. We note that the same kinetics for p53-RNASE4 complex formation were observed in IMR-90 cells. These results are presented in Fig. 9c.

Overall, with these extended studies of the kinetics and the biological relevance of ANG and RNASE4 at different stages after DNA damage the novelty and importance of our study have been expanded upon.

- 3) It is a little confusing that the authors begin the study by talking about super-enhancers, but the manuscript focuses on mRNAs differentially expressed in different senescence models, regardless of how they came to be elevated, something that other labs have also done. Presumably the authors have addressed these points on super-enhancer-driven transcription in the manuscript listed as reference 42, which did not appear to be provided in the submitted materials.

Response:

The discovery of novel genes with essential function to (senescent) cell characteristics is often approached from transcriptional screenings as the reviewer pointed out. However, such transcriptional profiling typically identifies hundreds to thousands of differentially expressed genes in senescent cells, which is overwhelming in terms of cell biological and biochemical follow-up experiments. By combining transcriptomics with the original super-enhancer concept, that super-enhancer associated genes control particularly important traits of cells (Hnisz *et al.* 2013 PMID

24119843), our rationale was to narrow the pool of transcriptionally activated genes in senescent cells to the more relevant ones so that it would be manageable to conduct cell biological and biochemical follow-up experiments that would advance our understanding of senescent cell biology.

In a recently published proof-of-concept study (Sturmlechner *et al.*, 2021, PMID 34709885), we screened for SASE-controlled genes that are highly conserved across species, cell types, and senescence-inducing stressors, which led to the identification of three such genes, *Hivep2*, *Rtn2* and *p21 (Cdkn2a)*. In-depth characterization of *p21* revealed that this p53 target gene activates retinoblastoma protein (Rb)-dependent transcription of at select gene promoters to produce a bioactive secretome, referred to as the P21-Associated Secretory Phenotype (PASP), that places stressed cells under immunosurveillance as part of a biological timer mechanism that controls cell fate.

In the current study, we further pursued the idea that SASE genes have particularly important functions in senescent cells but now relaxing our screening criteria to SASE-controlled genes conserved in MEFs (so only one cell type and one species) across three distinct senescence-inducing stressors (high-dose γ -irradiation, extensive replication or oncogenic signaling by KRAS^{G12V}). This yielded a manageable set of 50 genes, 28 of which we systematically characterized for relevant to the three core properties of senescent cells: arrest, survival under duress and production of a bioactive secretome. We then went on to validate whether the new insights obtained would be relevant to human cells if a SASE gene in MEFs was not conserved in IMR-90 cells but still transcriptionally more active in senescent cells through another (non-SASE) mechanism (as was often the case).

Overall, our approach is different for others that have studied super-enhancers in the context of senescence, but it turned out to be insightful and should be very useful in other contexts where transcriptomic information alone is too overwhelming for conducting cell biological and biochemical follow up studies.

Reviewer #2:

Sturmlechner et al. present a manuscript in which they identify a panel of candidate senescence-associated super-enhancer genes (SASEs) following H3K27Ac-chromatin immunoprecipitation followed by sequencing (ChIPseq). To identify the SASEs, the authors employ the following models of MEF senescence: high-dose γ -irradiation (IR), extensive replication (REP) or oncogenic signaling by KRASG12V (OI), and in the later stages of the paper human IR-senescent IMR-90 cells. The papers primarily focuses on RNase4 and ANG. shRNA mediated knockdown of these proposed SASEs does not enable resumption of the cell cycle in senescent cells but results in reduced cell survival and increased TUNEL +ve cells in senescence versus control cells. shRNA KD of RNase4 (but not ANG) results in the significant reduction in at least 1 out of 4 SASP genes tested. The reduced cell survival following KD of RNase4 or ANG can be rescued to some extent by concomitant p53 KD. RNase4 and ANG are shown to co-IP with p53.

The authors then explore the temporal kinetics of p53 (and p53S18) protein expression following irradiation, and show a convincing initial increase followed by a gradual decline in both up to a 15 day time point, and present data to uncouple p21 protein levels from p53. KD of ANG results in a significant increase of pro-apoptotic p53 targets (Puma, Noxa, Bax, Killer) and p21 (the later is

confusing), whilst RNASE4 KD significantly decreases Killer (and is thus proposed to act via p53, independent of its transcriptional activity).

RNASE4 is shown to be transcriptionally upregulated in IR-IMR-90s, whilst ANG is not. KD of RNASE4 in IR-IMR90s reduced cell survival, and similar temporal kinetics for p53 following senescence onset are demonstrated.

The manuscript is a very clearly written, with a solid logic flow, generally sound statements, although some of the conclusions could be tempered slightly. The presentation of the SASEs is novel and interesting. However, my overall impression of the manuscript is that the findings are not fully explored to warrant publication in a journal of this caliber. Detailed below are some suggestions that would help improve the manuscript further.

We are very pleased to hear that the reviewer felt the study was interesting, clearly written, and logically presented. We are very grateful for the reviewer's thoughtful and incisive comments and suggestions which were instrumental in further advancing the mechanistic depth and novelty of our study.

Suggested revisions:

1. Whilst RNAseq data indicates that expression of RNASE4 and ANG is upregulated in multiple models of MEF senescence, no evidence is provided that this results in an increase in protein expression. It will be important for the author to show that this occurs as part of their model. What happens to the levels of RNASE4 and ANG over a similar time course to that presented for p53?

Response:

This is a terrific point that aligns with point 2 of reviewer 1. Briefly, we were set back by the fact that we were unable to get the commercially available antibodies to work for immunoblotting. We decided to generate our own rabbit polyclonal RNASE4 antibody. Subsequent use of the RNASE4 antibody revealed that basal RNASE4 levels in proliferating MEFs are very low. However, RNASE4 levels substantially increase following DNA damage (prior to senescence), and then even further when progressing to senescence. These data have been incorporated in Fig. 6e, a figure that further contains related data that we obtained from addressing points 4 and 5 of the reviewer.

We note that increased transcription may be important to offset post-transcriptional mechanisms driving protein degrading when cell states change, and therefore higher mRNA levels may not be associated with an increase in protein levels *per se*.

2. The authors could additionally explore whether overexpression of RNase4 and ANG is able to induce senescence, or not, using existing constructs.

Response:

As suggested by the reviewer, we overexpressed Myc-Flag-RNASE4 or Myc-Flag-ANG in MEFs and examined the impact on rates of cell proliferation, apoptosis and senescence. We

found no or little impact on either of these rates both in the presence or absence of DNA damage. The data of these extended experiments are presented in Supplementary Figs. 5, 8, and 9a-d.

In contrast, in complementary experiments in which we knocked down *Rnase4* or *Ang* in WT MEFs and found that depletion of either protein has significant impacts on cell proliferation, apoptosis and senescence rates. Please see how we addressed points 4 and 5.

The manuscript would be further strengthened by experiments to explore whether p53 co-localises with either of these proteins in the early stages of senescence (e.g. PLA).

Response:

We are grateful to the reviewer for proposing these experiments. They turned out to be very insightful. Because suitable RNASE4 and ANG antibodies for immunostaining were not available, we stably expressed Myc-Flag-RNASE4 and Myc-Flag-ANG to conduct these experiments. We examined p53-RNASE4 and p53-ANG complex formation in primary MEFs 4 days after (when cells are not yet senescent) and 10 days after (when cells are senescent). The results of these experiments are shown in Fig. 7c-d and Supplementary Fig. 7a. They show that both ANG and RNASE4 indeed interact with p53 in the cytoplasm in a DNA damage-dependent fashion. This occurs prior to senescence, but the abundance of p53-RNASE4 and p53-ANG complex formation increases in the senescent state. We extended these studies to IMR-90 cells, focusing on RNASE4 because, as in MEFs, its expression is elevated upon senescence in these human cells. The results in IMR-90 cells were identical to those in MEFs. Please see the results shown in Fig. 9c. We further confirmed the presence of p53-RNASE4 and p53-ANG complex formation in the cytoplasm of senescent cells using a co-immunoprecipitation approach (see Supplementary Fig. 7b). page 10 (lines 1-11).

3. Can the authors provide evidence that RNASE4 is also a SASE in the IMR90s? At present only expression data is provided.

Response:

As noted in our initial submission, the conservation between mouse SASE and human SASE is very limited. From the 28 studied mouse SASE genes only 4 also acquire a SASE in IMR-90 cells. While MDM2 is a conserved SASE gene, RNASE4 is transcriptionally induced without SASE acquisition indicating the existence of SASE-independent mechanisms that support *RNASE4* transcription in senescent IMR-90 cells. Thus, the main idea is that SASE genes in one cell type and species may not be conserved as such in another cell type or species, but still upregulated for the same purpose via an alternative transcriptional mechanism (for instance via a common enhancer or a particular transcription factor, etc).

4. The paper diverges when the authors suggest that ANG promotes viability by inhibiting the transcription of pro-apoptotic p53 target genes, whilst RNASE4 acts independently of p53 transcription activity. The authors should consider exploring one of the angles in further details to provide more insight into the role of these SASEs in restraining senescent cell apoptosis.

Response:

To address this point, we conducted a series of experiments to deepen our understanding of how RNASE4 and ANG regulate p53 in response to stress cell fates and senescence in more detail.

Using immunofluorescence and subcellular fractionation, we found that in cycling cells and shortly after DNA damage (4 days post-irradiation), exogenous RNASE4 and ANG primarily localized to the cytoplasm including in the soluble cytoplasmic fraction and at membranes. Neither RNASE4 nor ANG overtly localized to mitochondria that we visualized with COX IV. On day 10 after irradiation when cells entered senescence, both RNASE4 and ANG also locate to the nucleus. A cytoplasmic pool of p53 without overt membrane localization was consistently detected in stressed cells at day 4 and to a lesser extent in senescent cells at day 10. Confirming earlier results, overall p53 levels were reduced in senescent cells at day 10 post-irradiation but residual p53 was mostly located in the nucleus. The overlapping localization of RNASE4, ANG and p53 in the cytoplasm throughout the time course and their consistent nuclear localization in senescent cells, opens the possibility that RNASE4 and ANG bind and suppress p53 activity both in the soluble cytoplasm and nucleus. The aforementioned PLA results (point #2) confirmed that RNASE4 and ANG interact with p53 predominantly in the cytoplasm, and to a lesser extent also in the nucleus of senescent cells. Please, find the subcellular fractionation results in Fig. 7b. Immunofluorescence results are presented in Supplementary Figs. 6 (for MEFs) and 10b (for IMR-90 cells). See also our description in the text on page 9 (lines 11-25).

We additionally found that RNASE4 or ANG are modulating cell fate decisions after DNA damage from cell death and towards cellular senescence and that over-expression of RNASE4 or ANG in irradiated cells counteracted cell death. For more details, please see the next point (point #5).

5. Are RNASE4 and ANG drivers of senescence (i.e. essential for restraining an apoptotic fate) or are the passengers in the establishment of the phenotype that function to restrain apoptosis following senescence establishment?

Response:

Another terrific point! The observation that RNASE4 and ANG promote survival of SNCs by limiting p53 activity, prompted the question as to whether they might also do so at earlier stages of the DNA damage response when cells are considered pre-senescent. To address this question, we depleted *Rnase4* or *Ang* from MEFs activated p53 with 4 or 10 Gy ionizing radiation, and measured cell cycle arrest, cell death and cellular senescence at 4 days post-irradiation (Fig. 6a). We found that *Rnase4* and *Ang* deficiency decreased cell proliferation at 4 Gy, while increasing apoptosis at 4 and 10 Gy and senescence at 10 Gy (Fig. 6b-d). As mentioned in our response to point 1, we complemented these studies with western blot analyses for RNASE4 to determine whether its levels increase in response to DNA damage. Indeed, RNASE4 was induced as early as 4 days after irradiation and further increased until day 10 post-irradiation when cells were considered senescent (Fig. 6e). See text on pages 8 (lines 23-26) and 9 (lines 1-8).

Collectively, these data suggest that RNASE and ANG are part of the cellular response to DNA damage to attenuate p53-mediated cell fate responses.

In other extended studies, we showed that RNASE4 or ANG overexpression (OE) had no impact on pre-senescent cell viability after DNA damage (Supplementary Fig. 8a). The same was

true for cell cycle arrest and entry into senescence (Supplementary Fig. 8b and c). On the other hand, overexpression of either protein did increase survival once cells had become senescent (Supplementary Fig. 8a). Furthermore, in the absence of DNA damage, RNASE4 or ANG OE had no discernable impact on MEF proliferation, survival or senescence rates (Supplementary Fig. 9a-d). Conversely, cell cycle arrest, apoptosis and cellular senescence were all markedly increased when RNASE4 or ANG was depleted instead of overexpressed in these experiments (Supplementary Fig. 9e-g). See text page 10 (13-20).

Collectively, these results suggest that RNASE4 and ANG restrain apoptosis and senescence when cells experience genotoxic stress, and apoptosis when cells are in a state of senescence.

6. As currently presented, the authors do not explore the SE feature at all in this manuscript. For example, what would mutation of the candidate SE mean for the establishment and maintenance of senescence?

Response:

Super-enhancer studies typically fall into two categories: one that focuses on the properties of the super-enhancers themselves (such as their activity, bound transcription factors, chromosome looping etc.) and one in which super-enhancers are used as a tool to identify target genes with critical roles in cell identity and characteristics.

The former approach to study super-enhancers in senescent cells has already been applied by Tasdemir *et al.* (2016) PMID 27099234 and Sen *et al.* (2019) PMID 30773298. In our study, we applied the second approach to discover novel genes for senescent cell maintenance.

We believe that there may be more insights to be gained when studying senescence-associated super-enhancers *per se*, but this would be outside of the scope of the current study.

7. Of note, p53 has previously been linked to RNASE4 (PMID 23284306) and ANG (PMID 22266868) – the authors should explore this publication in their discussion.

Response:

In cancer cell lines, ANG binds p53 to promote MDM2-mediated degradation of p53 and inhibit transcription of p53 target genes, including pro-apoptotic target genes, thereby increasing cancer cell survival. All this has been reported in the paper mentioned by the reviewer, Sadagopan *et al.* 2012 (PMID 22266868). We had referenced this paper, and it was not our intention to claim that ANG is a novel regulator of p53 *per se*. However, a p53-dependent role for ANG in senescent cells or the senescence program at large has not been reported. Furthermore, we have now included extended studies addressing the role of ANG in early fate decisions (prior to entry into senescence) in response to DNA damage, providing important new insights into its biological relevance and mechanism of action.

Concerning RNASE4, there is to our knowledge no literature on its role as a regulator of p53. The study cited by the reviewer (Xie *et al.* 2013, PMID 23284306) involves an shRNA-based genetic screen to identify genes that are preferentially required for proliferation of p53-deficient HCT116 cells relative to p53-sufficient HCT116 cells. *RNASE4* was identified as one of 103 such

genes identified in the screen. How RNASE4 depletion promotes proliferation of p53-deficient cells and the nature of the RNASE4-P53 relationship were not addressed.

Furthermore, we have extended our molecular-mechanistic work on RNASE4 and ANG in the context of senescence and cell fate decisions prior to senescence, which further adds to the overall novelty and importance of our study. Also, we have modified the discussion to better articulate the novelty of our initial findings and to include the added novelty of our extended studies. See revised manuscript page 13 (lines 23-26) and page 14 (lines 1-13).

8. It is interesting that both RNASE4 and ANG (RNASE5) are members of the RNase A superfamily. What happens to the other eight canonical RNases in IR-IMR90s? For example, is another family member able to act in a similar fashion to murine ANG? Could this provide a route into a more complete picture?

Response:

We thank the reviewer for this suggestion. We have checked the expression of other RNase A family members, *RNASE1*, *RNASE2*, *RNASE3*, *RNASE6*, *RNASE7* and *RNASE8* in RNA-seq data of proliferating and senescent IMR-90 cells. We found that besides *RNASE4* and *ANG*, none of the other family members are detected in our RNA-seq data suggesting that only *RNASE4* and *ANG* are expressed in IMR-90 cells.

Reviewer #3:

In this manuscript the authors investigate and present the important role of SASE target genes in fundamental properties of senescent cells. They find that a large number of SASE target genes that are activated upon induction of senescence, regardless of the inducer, promote the expression of SASP proteins and regulate the survival of senescent cells by suppressing p53-mediated apoptosis. Their findings are very interesting as they shed light on the complicated landscape of senescence.

Overall, this is a very good and promising piece of work that could be accepted for publication as long as the authors address some important issues mentioned below.

We thank the reviewer for the positive feedback and were encouraged by the fact that the reviewer found our study interesting and promising. As detailed below, we have worked to address all points raised by the reviewer.

Major points:

1. Experimental validation of the senescence phenotype of IR-, REP-, and OI-senescent MEFs should be provided.

Response:

We agree that this point is essential for our study, and we would like to refer to our recently published article (Sturmlechner *et al.*, 2021, PMID 34709885). Experiments validating our senescence models for IR-, REP- and OI-senescent MEFs and IR-senescent IMR-90 cells are

presented in Supplementary Figs. 1 and 4 of the article. We included assessments on cell cycle arrest, DNA damage markers, senescence-associated β -galactosidase assay, and thorough characterization of the senescence-associated secretory phenotype. We have included this reference at the beginning of our results section (page 5, lines 5-6), as well as appropriate method descriptions (page 16, lines 5-10).

2. In figure 2a and in supplementary figure 4a, the images are too bright.

Response:

We apologize for this technical issue. We believe that the brightness of images may have been changed during the manuscript upload process. We have now improved the brightness and contrast of these images.

3. In supplementary figure 2 it is important to show the value of shScr control.

Response:

As requested, we have included the shScr control bars for each candidate gene and shRNA in Supplementary Fig. 2. The data now illustrate the spread of the shRNA control group.

4. Western blotting would be better supported by an actual loading control marker such as GAPDH, actin etc. rather than Ponceau S staining.

Response:

Using single proteins as loading control for western blots with senescent cell lysates would have important caveats that can be circumvented by using Ponceau S instead. In our initial RNA-seq data, we found that conventionally used loading control proteins such as β -Actin, Tubulin or GAPDH are significantly and highly deregulated in senescent cells. Please, see the table below.

	IR-senescent versus proliferating		REP-senescent versus proliferating		OI-senescent versus proliferating	
	log2 fold change	p-value (adj)	log2 fold change	p-value (adj)	log2 fold change	p-value (adj)
Actb	-0.59	3.53E-11	-1.78	5.19E-57	-1.21	1.68E-05
Tuba1a	-0.72	3.05E-18	-0.94	4.92E-12	-0.94	0.008665
Tuba1b	-1.16	1.33E-20	-1.97	1.37E-42	-1.19	0.000289
Gapdh	-0.75	3.75E-07	-2.34	1.80E-49	-0.61	0.000761

We believe that Ponceau S staining which captures overall protein abundance is more appropriate to evaluate protein loading in experiments involving senescent cells. Additionally, we have used Ponceau S staining as loading control for many years and in publications also outside the senescence field. For example, please see, Limzerwala *et al* (2020) PMID 34841254, Sieben *et al* (2020) PMID 31738183, Aziz *et al.* (2018) PMID 30035751, Weaver *et al* (2016) PMID 27528194.

Minor points

1. In the following part of the manuscript text: “ These data are the first to indicate that the level of P53 activity that drives cell-cell arrest in response to genotoxic stress needs to be actively repressed upon entry into senescence in order to for SNCs to remain viable. By doing so, SNCs seemingly activate alternative mechanisms to sustain P21 levels that in volve SE formation” please replace “cell-cell arrest” with “cell cycle arrest”.

Response:

Thank you for pointing this mistake out. We have corrected it.

Reviewer #4:

This is a very interesting manuscript with potentially impactful results with respect to the understanding of epigenetic-molecular mechanisms driving senescence. The entire manuscript relies on the identification of SASEs and SASE-associated genes, and thus the analysis that is shown in supplemental Fig. S1 is of paramount importance to the rest of this body of work.

We appreciate the reviewer's positive feedback on our study and the helpful comments that helped us further improve the manuscript.

There are concerns with some of the methodology, including the identification of SEs and assigned genes that should be addressed below:

1. Supplemental Fig. 1a. Controls appear appropriate to eliminate genes responsive to non-senescence-induced stimuli.

Response:

Agreed. The experimental overview provided in Supplementary Fig. 1 will be helpful to readers to understand our experimental approach without having to work through all the details of our earlier paper (Sturmlechner *et al.*, 2021, PMID 34709885). We have further improved this overview by adding the number of replicates used in each group into the legend and method section (see also point 2).

We note that the way our super-enhancer data were used in the two studies is different and, in each case, led to unique new insights into the properties and vulnerabilities of senescent cells. We have emphasized this in the current paper (please see page 4 lines 11-23).

In the initial study, we applied a high stringency in that we focused on SASEs conserved in fibroblasts regardless of senescence-inducing stressors (REP, IR, OI) and species (mouse and human). We obtained very few common SASEs and followed up in-depth on the biological relevance of one of the associated genes, *p21*.

In the current study, we find that by limiting the comparative analysis to one cell type and one species and by introducing only one variable (distinct senescence-inducing stressors), important new general information about SNCs can still be obtained from common SASEs. This is because the genes they target in senescent MEFs are frequently activated through alternative mechanisms in human SNCs.

2. Materials and Methods, p19: How many biologic replicates were performed per condition for ChIP-seq? It is standard to perform at least two replicates. If that was not done, at least another replicate should be performed.

Response:

We thank the reviewer for this important point. All experimental groups for ChIP-seq consisted of two or three replicates and all groups for RNA-seq consisted for three replicates. This information was in Sturmlechner *et al.*, 2021 (see below), but has now been included in the legend of Supplementary 1a for convenience and in the method section (page 22, lines 3-6).

From Sturmlechner *et al.*, 2021:

“Generation of senescent and nonsenescent MEFs

For H3K27Ac-ChIP-seq experiments, two or three independent MEF lines were generated and induced to senesce via irradiation (IR), serial passaging (REP) or KRAS^{G12V}-overexpression (OI).”

“Generation of IR-senescent and control IMR-90 cells

H3K27Ac-ChIP-seq experiments and matched RNA-sequencing experiments were conducted in triplicate using three technical replicates.”

3. Materials and Methods, p20 and Suppl. Fig. 1b: regions defining SEs were excluded if they were within 2kb of any TSS, and yet in Suppl. Fig. 1b. every SASE is <2kb from the TSS of the three genes shown, and in fact covers the gene body or TSS of every gene shown. Please explain then why these are considered SASEs.

Response:

We agree that the described methodology and the H3K27Ac occupancy plots in Supplementary Fig. 1b are at first glance counterintuitive. In our approach, we have adhered to a commonly used methodology to stitch individual enhancers within 12.5 kb and exclude regions within +/- 2 kb from the TSS for super-enhancer discovery by ROSE. The methodology is now published in detail in Sturmlechner *et al* (2021) to which we refer to in this manuscript, including in the method section page 22, lines 2-6.

The existing literature recognizes that super-enhancers marked by H3K27Ac, MED1 or BRD4 tend to spread over the TSS and/or gene body, as reported by Richard Young’s group and others, for example in Loven *et al.* 2013 PMID 23582323 or Suzuki *et al.* 2017 PMID 28283057.

That being said, it is possible that super-enhancers spanning over the TSS are discovered by ROSE when the super-enhancer consists of peaks on both sides of the TSS. Despite the exclusion of +/- 2 kb of the TSS, the 12.5 kb stitching can join such peaks and make it appear that the TSS is part of the super-enhancer.

4. p20, identification of SEs using methodology of Whyte et al., 2013 (Cell): Unfortunately, H3K27ac ChIP-seq alone cannot be used to identify a super-enhancer. Whyte et al., had determined that only 21% (155/725) "SE"s identified by this method (occupancy slope ≥ 1) by H3K27Ac ChIP, were identified as SEs by MED1 or transcription factor ChIP. In order for the authors to properly classify the 50 SASEs, they will need to confirm by MED1 ChIP.

Response:

The reviewer's point is well taken. Because another MED1-based ChIP-seq experiment is not feasible, we previously ensured that we clearly articulated our criteria for SE identification in Sturmlechner *et al.* (2021). We further included information that super-enhancer identification is based on H3K27Ac in Supplementary Figure 1a. We also note that super-enhancer discovery that is solely based on H3K27Ac is often used in the literature, for example, please see: Jia *et al.* 2021 PMID 33579893; Joo *et al.* 2019 PMID 30638865; Peeters *et al.* 2015 PMID 26387944 or Ing-Simmons *et al.* 2015 PMID 25677180.

5. suppl fig. 1a. and p21, assignment of SE-associated genes. 50kb seems to be an arbitrary distance to assign SASE-genes given that SEs can affect genes anywhere within a topological domain, which can be as large as 1Mb. Could the authors describe where they came up with +/-50kb distance to assign genes to super-enhancers? In the paper they cite (Whyte et al., 2013, Cell) enhancers and SEs were assigned to nearest gene, regardless of distance. They also found that 93% of SEs were found within the same topological domain, so this approach can be added to confirm gene assignments.

Response:

We acknowledge that identification of super-enhancers and their putative target genes can vary among publications. We have followed the methodology by Loven *et al.* 2013 PMID 23582323, another *Cell* publication by Richard Young's lab to assign super-enhancers to putative target genes. The authors explain that the 50 kb window is based on previous literature that identifies "a large proportion of true enhancer/promoter interactions".

The 50 kb window is supported by Chepelev *et al.* 2012 PMID 22270183, who describes that most enhancer-promoter interactions occur within a distance of ~50 kb; and Dixon *et al.* 2012 PMID 22495300, who confirms that a large proportion of enhancer/promoter interactions happen within this window (and fall into topological domains).

The assignment of super-enhancers to genes within a window of up to +/- 50 kb from the TSS is commonly used in the existing literature besides Loven *et al.*, for example, please see: Jia *et al.* 2021 PMID 33579893; Khan *et al.* 2018 PMID 30169995, Ing-Simmons *et al.* 2015 PMID 25677180. Additionally, the super-enhancer database dbSUPER (developed by Khan *et al.* 2015 PMID 26438538) also uses the +/- 50 kb distance to identify putative super-enhancer target genes.

6. Materials and Methods, p22: How many biologic replicates were performed per condition for RNA-seq? It is standard to perform at least two replicates. If that was not done, at least another replicate should be performed.

Response:

Please see how we have addressed this comment in our response to points 1 and 2.

REVIEWERS' COMMENTS

Reviewer #1 (Remarks to the Author):

The authors have addressed my concerns in full.

Reviewer #2 (Remarks to the Author):

Sturmlechner et al. provide a substantially reviewed manuscript that includes additional figures, experimentally addresses many of my queries from the first submission, together with those of the other reviewers. As with the original manuscript, the revised manuscript and rebuttal letter is very clearly written, with solid statements throughout. The authors have diligently performed extensive work to revise the manuscript. Accordingly, I would like to thank and congratulate the authors for providing an expanded and more rounded manuscript. As a result, the manuscript's originally intriguing findings are further substantiated whilst still remaining timely and of great interest to the field.

I have just one point for clarification. The authors present new data in Fig 6, in which they explore the impact of RNASE4 or ANG knockdown followed by IR-induced senescence. They observe that this knockdown "decreased cell proliferation at 4 Gy, while increasing apoptosis at 4 and 10 Gy and senescence at 10 Gy (Fig. 6b-d)." They then go on to conclude that "Collectively, these data suggest that RNASE and ANG are part of the cellular response to DNA damage to attenuate p53-mediated cell fate responses."

However, in Supplementary Figure 9, the authors find that "cell cycle arrest, apoptosis and cellular senescence were all markedly increased when Rnase4 or Ang were depleted instead of overexpressed in these experiments (Supplementary Fig. 9e-g)." There seems to be a disconnect here, between these findings, which the authors should explore within the discussion.

Minor points

To avoid potential confusion by the reader, can the authors check their use of 'and' and 'or', especially for knockdown experiments. i.e. all shRNA work was down for individual targets, but the text can sometimes read as if there was a double knockdown.

Please add details for what is shown on the new western blots, in particular n number. Are these representative blots of independent experiments?

Congratulations of a lovely piece of work.

Reviewer #3 (Remarks to the Author):

The authors have properly addressed all issues raised and have improved significantly their manuscript rendering it appropriate for publication.

Reviewer #4 (Remarks to the Author):

Overall, the authors have addressed my concerns adequately, with one comment below. Thank you. p15 of rebuttal, response to reviewer #4 question 4: Given that it has been clearly established that H3K27ac alone cannot be used to definitively identify a super-enhancer (Whyte et al., 2013), the existence of published literature from this group and others suggesting the contrary without basis

does not satisfactorily address using this approach to identify SASEs. That being said, genes identified using this approach do appear to be relevant to senescence and I suppose there's no reason to be a stickler about this. I do think the authors should include a statement in the body of the results that their approach to identify SASEs is a simplified and less stringent approach to the established method of doing so.

RESPONSE TO THE COMMENTS OF THE REVIEWERS

We thank the reviewers for evaluating the revised version of our manuscript. We have addressed the few remaining comments as detailed below.

Changes to the text have been marked in blue font.

Reviewer #2 (Remarks to the Author):

Sturmlechner et al. provide a substantially reviewed manuscript that includes additional figures, experimentally addresses many of my queries from the first submission, together with those of the other reviewers. As with the original manuscript, the revised manuscript and rebuttal letter is very clearly written, with solid statements throughout. The authors have diligently performed extensive work to revise the manuscript. Accordingly, I would like to thank and congratulate the authors for providing an expanded and more rounded manuscript. As a result, the manuscript's originally intriguing findings are further substantiated whilst still remaining timely and of great interest to the field.

I have just one point for clarification. The authors present new data in Fig 6, in which they explore the impact of RNASE4 or ANG knockdown followed by IR-induced senescence. They observe that this knockdown “decreased cell proliferation at 4 Gy, while increasing apoptosis at 4 and 10 Gy and senescence at 10 Gy (Fig. 6b-d).” They then go on to conclude that “Collectively, these data suggest that RNASE and ANG are part of the cellular response to DNA damage to attenuate p53-mediated cell fate responses.”

However, in Supplementary Figure 9, the authors find that “cell cycle arrest, apoptosis and cellular senescence were all markedly increased when Rnase4 or Ang were depleted instead of overexpressed in these experiments (Supplementary Fig. 9e-g).” There seems to be a disconnect here, between these findings, which the authors should explore within the discussion.

Response: As requested, we have incorporated discussion of the outcomes of the above-mentioned experiments. Please see main text page 14, lines 15-21.

Minor points

To avoid potential confusion by the reader, can the authors check their use of ‘and’ and ‘or’, especially for knockdown experiments. i.e. all shRNA work was down for individual targets, but the text can sometimes read as if there was a double knockdown.

Response: Modified as requested throughout the manuscript.

Please add details for what is shown on the new western blots, in particular n number. Are these representative blots of independent experiments?

Response: Done as requested.

Reviewer #4 (Remarks to the Author):

Overall, the authors have addressed my concerns adequately, with one comment below. Thank you.

p15 of rebuttal, response to reviewer #4 question 4: Given that it has been clearly established that H3K27ac alone cannot be used to definitively identify a super-enhancer (Whyte et al., 2013), the existence of published literature from this group and others suggesting the contrary without basis does not satisfactorily address using this approach to identify SASEs. That being said, genes identified using this approach do appear to be relevant to senescence and I suppose there's no reason to be a stickler about this. I do think the authors should include a statement in the body of the results that their approach to identify SASEs is a simplified and less stringent approach to the established method of doing so.

Response: Modified as requested. Please see main text page 5, lines 5 to 8.